# The VelB IDD promotes selective heterodimer formation of velvet proteins for fungal development

Anna M Köhler[1,*] , Sabine Thieme[1,*] , Jennifer Gerke[1] , Karl G Thieme[1], Rebekka Harting[1], Kerstin Schmitt[1], Oliver Valerius[1], Wanping Chen[1] , Annalena Höfer[1], Emmanouil Bastakis[1], Anja Strohdiek[1], Emmanouil-Stavros Xylakis[1], Antje K Heinrich[2], Helge B Bode[2,3,4,5], Gerhard H Braus[1]

**Fungi possess several transcription factors with a characteristic velvet domain for DNA binding and homo- or heterodimerization, which is structurally similar to the mammalian NF-κB Rel homology domain. Velvet dimers control fungal development, virulence, and mycotoxin formation. VelB is the only regulator, which carries an intrinsically disordered domain (IDD) within the velvet domain. The IDD, as well as the positioning within VelB, is conserved in the fungal kingdom. Intrinsically disordered regions contribute to transcription activation and DNA binding and frequently appear in eukaryotic transcription factors. The VelB IDD provides selective heterodimerization and protein stability control. The IDD is not required for the formation of the VelB-VeA heterodimer of *Aspergillus nidulans* or *Verticillium dahliae*, but promotes the formation of the VelB-VosA heterodimer. The IDD destabilizes VelB single molecules and also balances its distribution and ratio between both velvet heterodimers. These balances contribute to control appropriate mycotoxin production and sexual development. Herewith, the VelB IDD represents a novel control mechanism of velvet protein stability and heterodimer formation for precise priming of fungal development.**

## Introduction

Intrinsically disordered proteins or domains possess an inherent flexibility with several possible conformations in solution instead of a well-defined 3D structure (1). This correlates with a significantly increased frequency of amino acid residues for higher net charge and lower mean hydrophobicity in comparison with ordered proteins (2). Eukaryotic DNA-binding proteins are significantly enriched in disordered domains (3). Posttranslational modifications of charged or other residues within these domains can alter functions or lifetime of the transcription factors by transient self-interactions or promiscuous binding to several partner molecules (4). Intrinsic disorder is present in transcriptional activation domains, which recruit the transcriptional machinery. Disordered domains can provide electrostatic interactions within dynamic complexes by following induced fit mechanisms for binding (4). A bioinformatics survey of human transcription factors revealed that DNA-binding domains with significant order are often flanked by regions with significant disorder (5).

The fungal Velvet-like B (VelB) transcription factor carries an intrinsically disordered domain within its DNA-binding and dimerization domain (6). VelB is one member of the conserved fungal velvet family of transcription factors, which also includes Velvet A (VeA), Viability of spores A (VosA), and Velvet-like C (VelC) (7). Velvet proteins control and coordinate development, virulence, and secondary metabolism including the formation of mycotoxins (7, 8, 9, 10, 11, 12, 13, 14). The velvet domain comprises ~100 to 200 amino acids and is a protein–protein interaction and DNA-binding domain with structural similarities to the Rel homology domain of the mammalian immune and infection response NF-κB regulator (6, 15). Velvet domain proteins and NF-κB regulators bind as homo- or heterodimers to a myriad of genomic sites. The NF-κB heterodimer p65p50 interface provides increased conformational plasticity resulting in significantly stronger affinity to DNA than the corresponding homodimers p50p50 or p65p65 (16).

Fungal VelB acts as light-dependent multifunctional regulator for fungal asexual and sexual development. VelB coordinates differentiation with the appropriate secondary metabolism and furthermore controls spore viability (17, 18). VelB can form a heterodimer either with VeA (VelB-VeA) or with VosA (VelB-VosA) (19). *Aspergillus* asexual development is promoted by light and results in the release and dispersal of spores (conidia) into the air (7, 20). VelB activates the *brlA* (*bristle A*) gene encoding the central regulator of the progression of conidiation (21, 22). In contrast,

---

[1]Molecular Microbiology and Genetics and Göttingen Center for Molecular Biosciences (GZMB), University of Göttingen, Göttingen, Germany   [2]Institute of Molecular Biosciences, Biocentre, Goethe-University Frankfurt, Frankfurt am Main, Germany   [3]Department of Natural Products in Organismic Interactions, Max-Planck-Institute for Terrestrial Microbiology, Marburg, Germany   [4]Center for Synthetic Microbiology (SYNMIKRO), Phillips University Marburg, Marburg, Germany   [5]Department of Chemistry, Phillips University Marburg, Marburg, Germany

Correspondence: gbraus@gwdg.de
*Anna M Köhler and Sabine Thieme contributed equally to this work

VosA represents a negative regulator of asexual development and represses the *brlA*-dependent genetic networks of asexual development, oxidative stress response, and the corresponding secondary metabolism (23, 24). *Aspergillus* develops closed sexual fruiting bodies in darkness and low oxygen pressure–representing overwintering structures of this mold in the soil. Specific multi-nuclear cells (Hülle cells) are formed to nurse the growing fruiting body and to protect it by mycotoxins against fungivores (19, 20, 25, 26). Nuclear VelB-VeA heterodimer activates sexual development. This transcriptional regulation is linked to epigenetic control by the formation of a trimeric complex of VelB-VeA with the methyltransferase LaeA (loss of the aflatoxin regulator expression A) as a global regulator of secondary metabolite formation. LaeA is required to synthesize the aflatoxin family mycotoxin sterigmatocystin for protection of the sexual fruiting bodies (18, 19, 27, 28). VelB-VosA is important for trehalose biogenesis to support spore viability and germination (19, 23).

The different VelB interactions are required in response to diverse environmental signals to support distinct developmental programs and the appropriate secondary metabolism (14, 18). The molecular mechanisms of how the fungal cell distributes VelB to the two alternative heterodimers to regulate different sets of target genes are yet unknown. VelB-VeA supports nuclear entry of VelB (18). This suggests that nuclear VelB-VosA formation might happen by an exchange of the VelB-binding partner after nuclear entry. This exchange of the VelB-binding partner presumably depends on external signals and the cell type–specific status of *Aspergillus nidulans* development. Specific control mechanisms for differential gene expression and protein stability regulation, which change the protein homeostasis of velvet domain proteins, are essential for heterodimer formation control (20, 29). Internal protein signal sequences within velvet domain proteins for the promotion of selective heterodimer formation are yet unknown.

We compared the VelB intrinsically disordered domain within the fungal kingdom and characterized and analyzed its molecular role in fungal cells or during development to gain further insights into the evolution and function of velvet domain regulatory proteins. The insertion in the VelB velvet domain is evolutionarily conserved in filamentous fungi of different divisions. The intrinsically disordered domain (hereafter IDD) has a significant impact on protein stability and an even more remarkable potential to select and control VelB protein interactions. The IDD is required to control cellular ratios of different VelB heterodimers and therefore links fungal development to the required secondary metabolite production.

## Results

### VelB is the only member of the velvet family carrying an intrinsically disordered region within the DNA-binding and dimerization domain

The crystal structures of the *A. nidulans* VosA and VelB velvet domains revealed a similar fold to the Rel homology domain of the mammalian transcription factor NF-κB (6). The amino acid sequence similarities between Rel homology and velvet domains comprise only ~14%, but important DNA contact sites are conserved. The VelB-VosA$_{1–190}$ heterodimer crystal structure lacks a 99–amino acid (aa) insertion within the VelB velvet domain, which has been removed because of protease treatment during crystallization (6). Sequence analysis has predicted that this 99-aa sequence is unstructured (Fig S1) and in the following is denominated as intrinsically disordered domain (IDD; Fig 1A). VelB is the only member of the *A. nidulans* velvet family with an IDD, which is absent in VeA, VelC, or VosA (14).

A comparison of the deduced VelB amino acid sequences of different fungal genomes provided by the JGI fungal genome database (30) revealed that the insertion of an IDD at a similar position into the velvet domain as in *A. nidulans* is conserved in numerous fungal VelB counterparts (Fig 1B). Deduced VelB IDD sequences of different species from the fungal kingdom (Ascomycota, Basidiomycota, Zygomycota, Chytridiomycota) share the presence of numerous serine residues, but differ considerably (Fig S2). There is a remarkable conservation of five-aa residues at the N-terminal IDD boundary inserted into the VelB velvet domain in all four divisions. This boundary is mostly terminated with a positively charged amino acid residue as arginine or lysine, followed by a histidine residue (red box in Figs 1B and S2A–D). Ascomycota share, adjacent to the conserved 5'IDD boundary, a serine-rich conserved 15-aa motif (Motif$_{IDD}$) at the N terminus of the IDD (turquoise box), which is absent in the IDDs of the VelB counterparts of Basidiomycota, Zygomycota, or Chytridiomycota (Figs 1 and S2A–D). The VelB velvet domain C-terminal boundary of the IDD is only conserved among ascomycetes (red box in Fig 1B) and includes a highly conserved central threonine residue, which is also found in the 3' boundary region of various basidiomycetes (Fig S2A–D).

The different fungal groups show strong variations in the size of the analyzed VelB IDDs. Ascomycota and Basidiomycota have larger average sizes of 105 and 204 aa, respectively, compared with the IDDs of Zygomycota or Chytridiomycota, which are significantly smaller (28 and 33 aa in average, Figs 1 and S3). Ascomycota or Basidiomycota normally possess one *velB* gene. Zygomycota or Chytridiomycota species often acquired two or more *velB*-like gene copies. The Zygomycetes carry genes for VelB with very short IDDs. Whereas the lack of the IDD (zero amino acids) was never observed, there are short Zygomycetes IDDs with six amino acids, which are very not conserved in their sequence in other fungi (Tables S1 and S2).

Taken together, the fungal VelB orthologs share a region of intrinsic disorder at the same position within the velvet domain and that differs in size and sequence with increasing evolutionary distance. These different IDD sequences and sizes combined with up to six *velB*-like genes present in fungal groups as the Chytridiomycota suggest a rapid *velB* gene evolution, including gene duplications and subsequent IDD variations. A common molecular IDD function is yet elusive, but these fungal VelB insertions presumably would have not been conserved during evolution if they had only occurred randomly or unspecifically.

### The intrinsically disordered domain destabilizes *A. nidulans* VelB

Control of the relative amounts of transcription factors is essential to ensure appropriate heterodimer formation for the regulation of target genes in response to changing environments. VelB full-length protein stability was compared with a variant lacking the

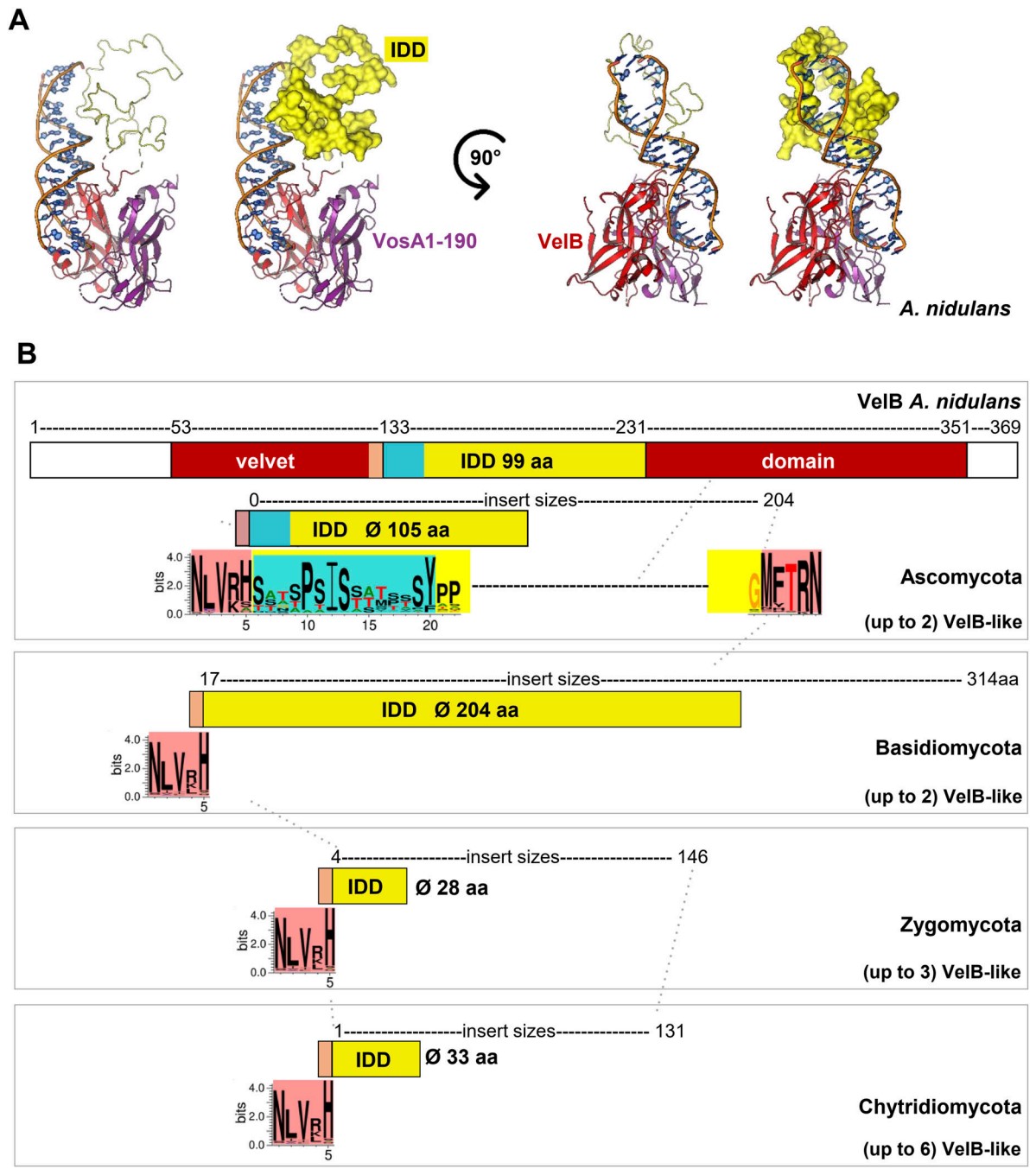

**Figure 1.   Genes encoding the VelB velvet transcription factors within the fungal kingdom share an intrinsically disordered domain, which is inserted at the same position within their DNA-binding and dimerization domain.**
**(A)** Crystal structure–derived model of the *A. nidulans* VelB (red)-VosA$_{1-190}$ (purple) heterodimer bound to DNA (according to reference 6; PDB 4N6R) lacking the VelB intrinsically disordered domain (IDD, 99 amino acids; Fig S1). The IDD was most likely cut by protease treatment during crystallization. The unstructured PhoA1–150 NMR conformer 6 (PDB 2MLY) has a similar size as the structurally unknown VelB IDD and was used as a template in a model to illustrate the size of the VelB IDD (yellow). **(B)** VelB velvet domain N-terminal boundary of the IDD is highly conserved in the fungal kingdom, and its C-terminal residue is in most cases a histidine. The *A. nidulans* VelB IDD of 99 amino acids includes a 15–amino acid motif (turquoise) and a 3′ boundary, which is conserved among Ascomycota. Averages of the amino acid sequence sizes of deduced VelB inserts differ among fungal divisions as indicated in the yellow boxes. The deduced IDD information and the corresponding genes are listed in Table S1 source data 1 for fungal species with single genes for VelB or in Table S2 for species with at least two isogenes.

IDD in fungal strains expressing the proteins C-terminally fused to GFP (VelB-GFP, VelB$^{\Delta IDD}$-GFP) under the control of the endogenous *velB* promoter. Therefore, *velB::sgfp* and *velB*$^{\Delta IDD}$*::sgfp* fusion constructs were transformed in the locus of the Δ*velB* strain. The fungal cultures were supplemented with cycloheximide after 24 h of vegetative growth to inhibit protein biosynthesis. Protein

crude extracts were analyzed by Western experiments hourly from 0 to 5 h after supplementation. Cycloheximide assays revealed that full-length VelB-GFP is less stable than VelB$^{\Delta IDD}$-GFP during filamentous growth. Whereas ~90% of VelB$^{\Delta IDD}$-GFP was still present after 5 h, the relative protein amounts of VelB-GFP with the IDD were reduced to ~56% (Fig 2A). This suggests that the IDD provides a destabilizing function, which allows the fungal cell to control VelB protein turnover under specific environmental conditions.

Relative VelB protein abundance in protein crude extracts of both versions was compared between vegetative growth and upon induction of asexual development when the VelB-VosA heterodimer is active. The relative normalized expression of *velB$^{\Delta IDD}$* compared with *velB* was similar from six to 18 h of incubation under asexual conditions (Fig S4). The transcript level of *velB* was higher compared with *velB$^{\Delta IDD}$* after 18 h of asexual development. This suggests that the IDD has no influence on the transcript level of *velB*. The constant level of *velB$^{\Delta IDD}$* transcripts resembles the protein amount of VelB$^{\Delta IDD}$-GFP (Figs 2B and S4).

Vegetative cultures incubated for 24 h comprise the 67-kD VelB-GFP protein (Fig 2B, red arrow) or the 56-kD VelB$^{\Delta IDD}$-GFP protein (blue arrow), respectively. Full-length VelB-GFP was only detectable during early asexual development but is mostly degraded after 18 h of differentiation when only free GFP is still present (~27 kD, Fig 2B, green arrow). This indicates a rapid VelB degradation during ongoing asexual development. In contrast, VelB$^{\Delta IDD}$-GFP fusion protein degradation is slower, with ~30% of the relative protein abundance present after 48-h incubation under asexual development–inducing conditions (Fig 2B). The additional ~55- and ~40-kD bands in the VelB-GFP Western experiment likely represent truncated or degraded forms of the protein. Mass spectrometry detected internal VelB peptides for the 55 kD band but lacked N- or C-terminal sequences, supporting this. The degradation of VelB-GFP and VelB$^{\Delta IDD}$-GFP does not affect the resulting free GFP. GFP has a stable protein structure, and no GFP-specific protein degradation pathways exist; therefore, its degradation occurs slowly. In addition, the VelB$^{\Delta IDD}$-GFP fusion protein showed a double band compared with the full-length VelB-GFP, suggesting different posttranslational modifications (PTMs) of VelB$^{\Delta IDD}$-GFP. The mass spectrometry indicated a phosphorylation site at threonine 84 (T84) specific to the VelB$^{\Delta IDD}$. The different posttranslational modification states of VelB$^{\Delta IDD}$ might result from the phosphorylation at T84.

This corroborates that in *A. nidulans*, the presence of the IDD in the WT VelB protein contributes to protein degradation, especially during light-induced asexual development. In contrast, truncated VelB without IDD is more stable under the same cultivation conditions, which supports the destabilizing impact of IDD on the VelB protein. In conclusion, these data suggest a VelB IDD-dependent degradation control as a mechanism to restrict cellular VelB abundance for its adjusted channeling to complex formation with either VeA or VosA during fungal development.

### The VelB intrinsically disordered domain enables selective heterodimer formation with VosA, which ensures nuclear localization of VelB

Independently of light, VelB is localized in the cytoplasm before its nuclear import and in the nucleus for DNA binding (18).

Fluorescence microcopy was applied to examine whether the IDD affects VelB nuclear localization. These microscopy experiments were carried out to investigate whether the IDD affects VelB localization in dependence of VeA or VosA. Strains with full-length VelB fused to GFP (VelB-GFP) were compared to ones with the truncated version without IDD (VelB$^{\Delta IDD}$-GFP). A constitutively expressing GFP (OE GFP) strain and the WT strain served as controls to exclude unspecific GFP background signals (Fig S5A). Independent from the presence of the IDD, VelB accumulated in nuclei of vegetative growing hyphae; however, an additional subpopulation appeared in the cytoplasm (Fig 3A). VelB-GFP and VelB$^{\Delta IDD}$-GFP localization was analyzed in *veA* and *vosA* single and double deletion strains after 18 h of vegetative growth. VelB-GFP is localized in the nuclei, whereas VelB$^{\Delta IDD}$-GFP is dispensed throughout the hyphae without nuclear accumulation in the absence of *veA* (Fig 3A). Nuclear VelB accumulation is not altered when *vosA* is missing, but both VelB variants are absent from nuclei in strains lacking *veA* and *vosA* (Fig 3A).

Intrinsically disordered proteins can function as hubs in protein interaction networks (44–46). VeA enhances the transport of VelB through the heterodimer VelB-VeA from the cytoplasm into the nucleus in the dark (18), whereas the VelB-VosA heterodimer resides predominantly in the nucleus (19).

GFP pull-down experiments were conducted to investigate whether VelB without the IDD still interacts with VeA or VosA within the vegetative fungal cell. Mycelium from vegetative cultures was used for GFP pull-downs followed by LC-MS analysis for the identification of interacting proteins. The bait proteins VelB-GFP and VelB$^{\Delta IDD}$-GFP were detected with similar LFQ (label-free quantification) intensities and MS/MS counts in biological replicates (Table 1). Comparable numbers of unique peptides were identified, indicating that the pull-downs worked equally well for both VelB variants. Proteins were filtered for detection in at least two out of three biological replicates with MS/MS counts ≥ 4, unique peptides ≥ 3, and logarithmic LFQ intensity ≥ 20, and were absent in the control strain (Tables 1 and S3).

The GFP pull-downs revealed 32 interaction candidates exclusively found for VelB$^{\Delta IDD}$-GFP, which were not detected with full-length VelB-GFP as bait (Fig S6). These proteins were sorted and grouped according to their known or predicted cellular function or localization: mRNA translation, primary metabolism, RNA maturation and processing, membrane/cell wall, signaling, cell compartments, DNA binding, and unknown function (Table S3, Fig S6A). For almost half of the putative interaction partners (14 proteins), a nuclear localization signal with a score ≤ 5 was predicted by employing the cNLS Mapper program ((34), Table S3, underlined AN numbers). Proteins with this score presumably can shuttle between, and can be localized in both the cytoplasm and the nucleus. The velvet domain protein VeA was pulled by both VelB-GFP protein variants with similar efficiency in all experiments. Similarly, the catalase CatB was also always identified as VelB-interacting protein supporting a link to the fungal oxidative stress response (Tables 1 and S3). Only one protein was exclusively pulled with VelB-GFP: the velvet domain protein VosA.

Co-immunoprecipitation (Co-IP) experiments from vegetative mycelium of strains expressing *veA:HA* or *vosA:HA* and *velB:gfp* or *velB$^{\Delta IDD}$:gfp* under their native promoters were conducted to verify

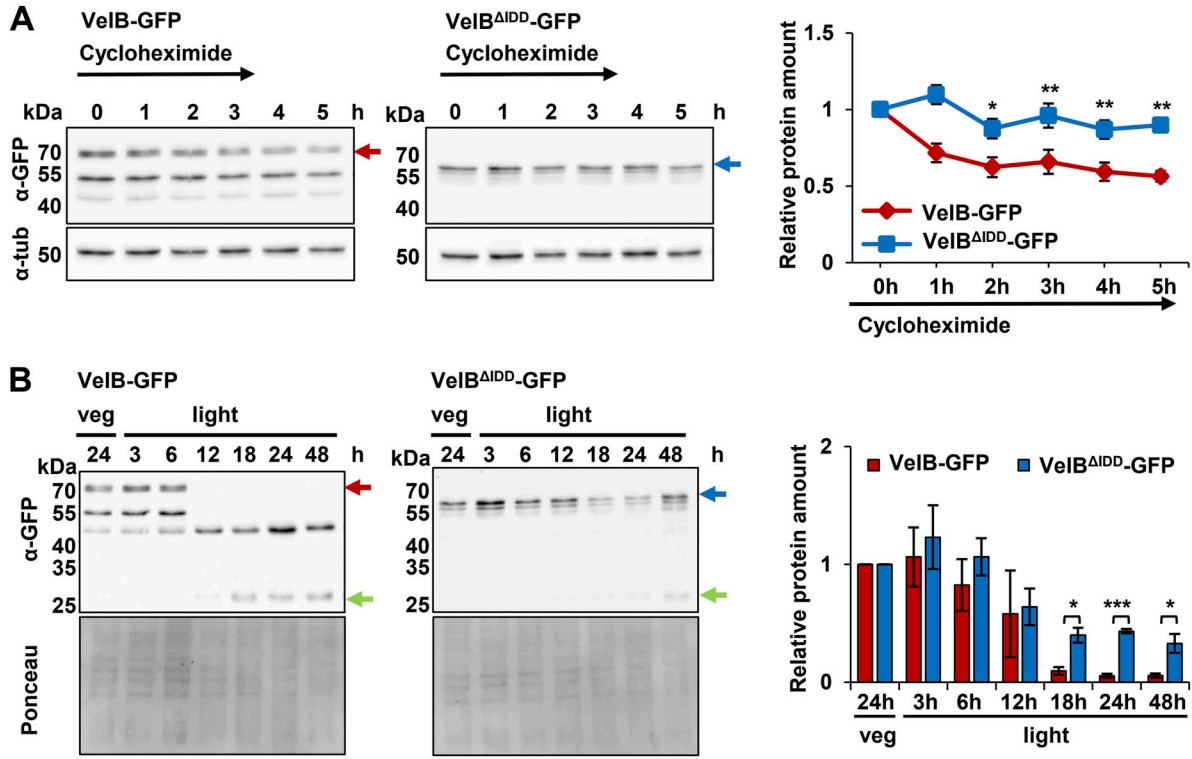

**Figure 2.   IDD destabilizes *A. nidulans* VelB.**
**(A)** Cycloheximide chase analysis of VelB-GFP and VelB^ΔIDD^-GFP protein degradation. Cycloheximide was added after 24 h of vegetative fungal growth at 37°C to submerged cultures. Samples were taken from zero to 5 h post-supplementation. Western experiments applying the α-GFP antibody to crude extracts show less stable VelB-GFP compared with VelB^ΔIDD^-GFP. α-Tubulin was used as a loading control. Lack of the IDD increases protein stability. Error bars indicate the SEM of four biological replicates normalized against the tubulin signal. *P*-value was calculated with SD. *P < 0.05, **P < 0.01. **(B)** Relative abundance of VelB-GFP or VelB^ΔIDD^-GFP during fungal development. Vegetative grown mycelia were shifted after 24 h to solid minimal medium and cultivated for indicated time periods in the light at 37°C for induction of asexual development. Western analysis shows that VelB-GFP (red arrow) is degraded during early asexual development when degradation products and free GFP (green arrow) become visible. VelB^ΔIDD^-GFP (blue arrow) is stable and still detectable after 48-h cultivation in the light. The diagram depicts the quantification of VelB-GFP and VelB^ΔIDD^-GFP relative to the protein amount of vegetatively grown cultures normalized against Ponceau. Error bars indicate the SEM of three biological replicates. *P*-value was calculated with SD. *P < 0.05; ***P < 0.005.

the velvet heterodimer formation of VelB with VeA and VosA. Therefore, the relative protein amount of the VelB-GFP and VelB^ΔIDD^-GFP was compared regarding the protein amount of the corresponding VeA-HA or VosA-HA proteins. Western experiments revealed that VeA-HA signals with equal intensity in both GFP pull-downs (Fig 3B, black arrow) and the VeA-HA pull-down resulted in comparable abundance of VelB-GFP (HA pull-down, red arrow) with or without IDD (HA pull-down, blue arrow). These experiments confirmed that VelB-VeA interaction is independent of the VelB IDD. The Co-IPs with VelB variants and VosA show VelB-GFP and VelB^ΔIDD^-GFP with similar abundance in the GFP pull-down experiments (Fig 3C, red/blue arrows). In contrast, VosA-HA was only detected for the full-length VelB-GFP pull-down (Fig 3C, GFP pull-down, violet arrow). VelB was detectable by label-free quantification, and TSIM was performed for peptides outside the IDD region. However, because of limited coverage within the IDD, these data cannot fully resolve IDD-dependent stability differences. Targeted analysis of the IDD region would be needed for confirmation. Quantification of Western blot signal intensities revealed a 10-fold increased abundance of VosA in the VelB-GFP strain compared with the VelB^ΔIDD^-GFP strain. The reciprocal experiment

resulted in the identification of VosA-HA in equal amounts for both HA pull-downs (Fig 3C, HA pull-down, violet arrow). VosA-HA co-enriched VelB-GFP (HA pull-down, red arrow) but never VelB^ΔIDD^-GFP. These findings corroborate that the IDD prevents multiple additional interactions, which were monitored in the VelB^ΔIDD^-GFP pull-down. The IDD specifically promotes the formation of the VelB-VosA heterodimer in *A. nidulans* in vivo during vegetative growth conditions.

Equivalent pull-downs were conducted with VelB orthologous proteins Vel2-GFP and Vel2^ΔIDD^-GFP of the plant pathogenic ascomycete *Verticillium dahliae* to investigate whether the involvement of the IDD for the interaction with VosA is conserved in different fungi. The interactome of *V. dahliae* Vel2^ΔIDD^-GFP showed a decrease in potential interaction partners compared with *A. nidulans* Vel2^ΔIDD^-GFP (Fig S6B and C and Table S4). *V. dahliae* Vel1 (VeA) can interact with Vel2 with or without IDD. In contrast and as found in *A. nidulans*, Vos1 as a counterpart of VosA can only significantly interact with the Vel2 full-length protein with the IDD (Fig S6B and C). These experiments revealed and support an evolutionarily conserved IDD function in heterodimer partner selection.

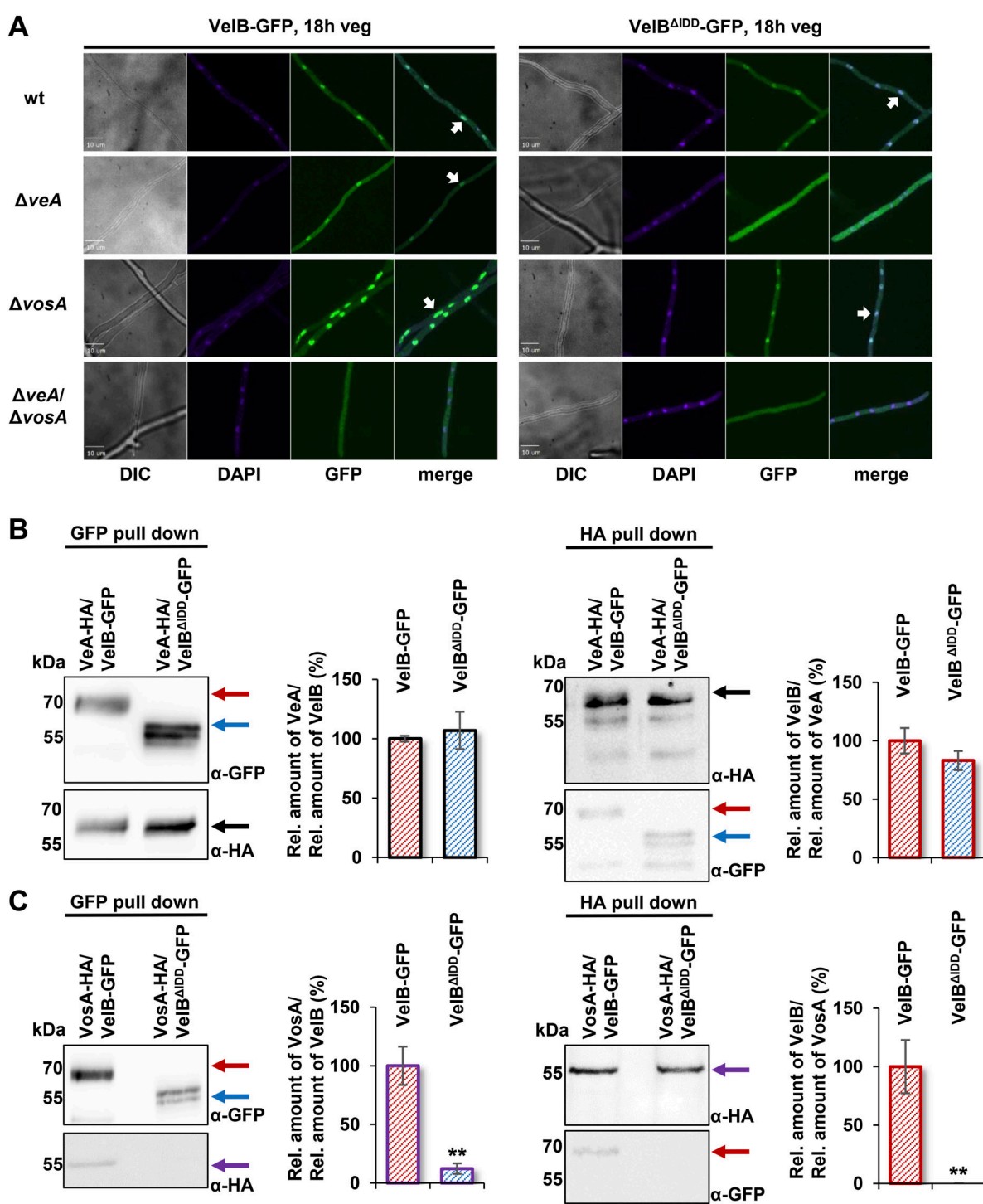

**Figure 3. *A. nidulans* VelB IDD is required for VelB-VosA heterodimer formation but not for VelB-VeA dimer formation or VelB nuclear localization.**
**(A)** Fluorescence microscopy of VelB-GFP with or without IDD revealed nuclear accumulation (white arrows). *A. nidulans* strains were grown vegetatively in submerged cultures for 18 h at 30°C. Predominant nuclear localization of VelB-GFP is similar in wt, *veA* (Δ*veA*), and *vosA* (Δ*vosA*) single deletion strains, but there are changes in cytoplasmic localization in the veA/vosA double deletion strain (Δ*veA*/Δ*vosA*). VelB^ΔIDD-GFP is localized in the nucleus in wt and *vosA* deletion strains but found in the cytoplasm in *veA* or *vosA*/*veA* deletion strains. Nuclei were visualized with DAPI. wt, wild type; DIC, differential interference contrast; scale bar = 10 μm. **(B)** Co-immunoprecipitation experiments included GFP and HA pull-downs of strains expressing either VeA-HA, VelB-GFP, or VelB^ΔIDD-GFP fusion proteins followed by Western experiment detection with GFP and HA antibodies. The relative amount of VeA against VelB (black diagram) or the relative amount of VelB against VeA (red diagram) was quantified and revealed that VelB and VeA form a heterodimeric velvet complex with and without the IDD. **(C)** Co-immunoprecipitation experiments of strains expressing VosA-HA and VelB-GFP or VelB^ΔIDD-GFP fusion proteins were followed by Western experiment detection with GFP and HA antibodies. The relative amount of VosA against VelB (violet diagram) or the relative amount of VelB against VosA (red diagram) was quantified, revealing that VelB without IDD did not form a VelB-VosA complex. Error bars indicate the SEM of three biological replicates. *P*-value was calculated with SD. **P < 0.01. Black arrows = VeA-HA; red arrows = VelB-GFP; blue arrows = VelB^ΔIDD-GFP; violet arrows = VosA-HA.

**Table 1. Velvet domain proteins identified from GFP pull-downs of VelB-GFP and VelB$^{\Delta IDD}$-GFP.**

| Sys. name | Std. name | LFQ intensity | | MS/MS counts | | Unique peptides | |
|---|---|---|---|---|---|---|---|
| | | *velB:gfp* | *velB$^{\Delta IDD}$:gfp* | *velB:gfp* | *velB$^{\Delta IDD}$:gfp* | *velB:gfp* | *velB$^{\Delta IDD}$:gfp* |
| AN0363 | VelB | 28.88 | 29.77 | 735 | 728 | 15 | 16 |
| | | 26.18 | 27.77 | 191 | 139 | 11 | 9 |
| | | 25.22 | 28.05 | 112 | 36 | 13 | 9 |
| AN1052 | VeA | 25.51 | 26.35 | 406 | 405 | 17 | 16 |
| | | 23.75 | 25.85 | 125 | 98 | 13 | 10 |
| | | 23.78 | 26.01 | 89 | 47 | 16 | 12 |
| AN1959 | VosA | 24.32 | 14.92 | 166 | 1 | 11 | 1 |
| | | 22.03 | 15.86 | 8 | 0 | 3 | 0 |
| | | 16.89 | 15.24 | 1 | 0 | 1 | 0 |
| AN0807 | LaeA | 21.35 | 20.68 | 47 | 30 | 7 | 6 |
| | | 17.39 | 16.38 | 0 | 0 | 0 | 0 |
| | | 19.74 | 17.22 | 2 | 0 | 2 | 0 |

VelB-GFP with and without the IDD reliably pulled the VeA protein. VosA was only identified in experiments with the full-length VelB-GFP but not with VelB$^{\Delta IDD}$-GFP. In contrast, the IDD is dispensable for VelB-VeA-LaeA velvet complex formation. The values for LFQ intensity, MS/MS counts, and unique peptides represent the mean value from three biological replicates. Sys. Name, systematic name; Std. name, standard name; MS/MS counts = number of specific peptides fragmented and analyzed. Descriptions were obtained and adapted from FungiDB, NCBI, and Ensembl Fungi (31, 32, 33).

Protein abundance was analyzed by Western experiments and shows similar protein levels for all strains investigated (Fig S5B). The same localizations were observed after 18-h sexual and asexual development (Fig S5A, C, and D). These results indicate that external factors like light or darkness do not influence VelB localization in the early developmental state. However, all these data support that VelB needs the interaction of either VeA or VosA for nuclear localization in all tested conditions.

### VeA counteracts VelB-VosA heterodimer formation and suggests VeA as preferred VelB interaction partner

VelB-VosA heterodimer formation according to the previously determined crystal structure was aimed for by the recombinant expression of the full-length VelB protein with a truncated version of VosA encompassing residues 1–190 (VosA$_{1-190}$) in *Escherichia coli* (6). Therefore, it was investigated whether a full-length VosA requires the VelB IDD for heterodimer formation in vitro. Full-length fusion proteins of VosA-GST and VelB-His or VelB$^{\Delta IDD}$-His were recombinantly expressed in *E. coli* followed by GST pull-downs where the lysate of VosA-GST was mixed with either purified VelB-His or VelB$^{\Delta IDD}$-His. VosA-GST (Figs S3 and S7, violet arrow) co-enriched with VelB-His (red arrow) and VelB$^{\Delta IDD}$-His (blue arrow). This result demonstrates that the VelB IDD itself is not directly required for the interaction of VelB with VosA in vitro. This supports indirect effects of other cellular components that impact the IDD to prevent the VelB-VosA interaction in vivo.

Fungal ΔvelB/ΔveA and velB$^{\Delta IDD}$/ΔveA strains were generated to analyze whether the VelB-VosA heterodimer formation is affected by VeA and the IDD in vivo. Samples of these strains were analyzed from vegetative mycelium and from light-induced asexually grown mycelium because the VelB-VosA heterodimer is a regulator of this developmental program (23). Western experiments show similar

protein intensities of VelB-GFP and VelB$^{\Delta IDD}$-GFP in the *veA* deletion strain from vegetative and asexual mycelium (Fig S8A). GFP pull-downs with VelB-GFP and VelB$^{\Delta IDD}$-GFP in *veA* deletion background were performed. In the absence of the *veA* gene, VosA is able to bind to VelB independently of the IDD (Fig S8B, Table S5). Co-IP experiments with VelB-GFP or VelB$^{\Delta IDD}$-GFP and VosA-HA in strains deleted in *veA* show a similar result. VosA-HA was able to recruit VelB$^{\Delta IDD}$-GFP and vice versa in Δ*veA* (Fig S9).

These data suggest that heterodimerization of VelB$^{\Delta IDD}$ with VosA is possible in the absence of *veA* in vivo and accordingly is also possible in vitro. The presence of an intact *veA* gene favors heterodimerization with VeA as a preferred interaction partner of VelB in vivo.

### The VelB IDD promotes asexual development in *A. nidulans*

The VelB-VeA and VelB-VosA heterodimers coordinate fungal development (18). VelB-VeA and VelB-VosA are required for the sexual and asexual pathway, respectively (18). The impact of the VelB IDD on *A. nidulans* developmental programs was examined by comparing phenotypes of WT, *velB* deletion (Δ*velB*), or *velB$^{\Delta IDD}$* mutant strains.

Light/illumination promotes asexual development and results in conidiophores with green asexual spores on agar plates for the WT and the Δ*velB* strain (Fig 4A). Point inoculations of the strains lead to similar colony morphology of the *velB$^{\Delta IDD}$* and the WT strain when incubated in asexual development–inducing conditions (Fig S10A). In contrast, spreading spores area-wide on culture plates resulted in a different phenotype. The strain lacking the VelB IDD forms increased amounts of aerial hyphae as precursors of conidiophores resulting in a white fluffy appearance (Fig 4A, red arrow). Conidiophores with green conidiospores are rare and only formed at the edge of the agar plate (Fig 4A, blue arrow).

# Life Science Alliance

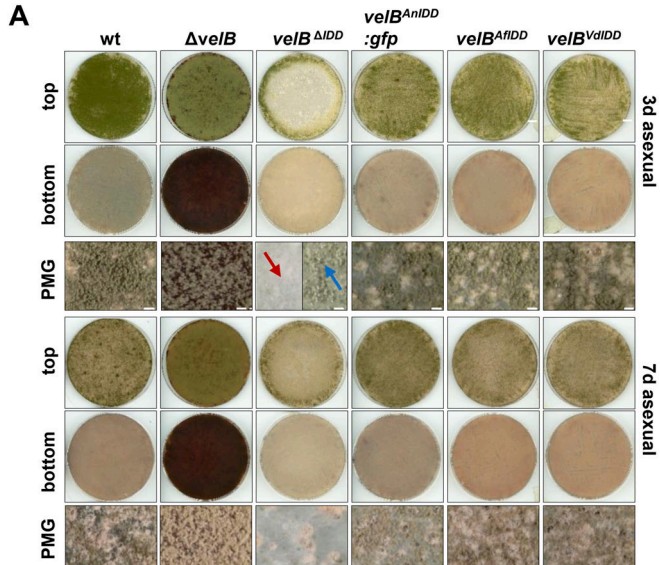

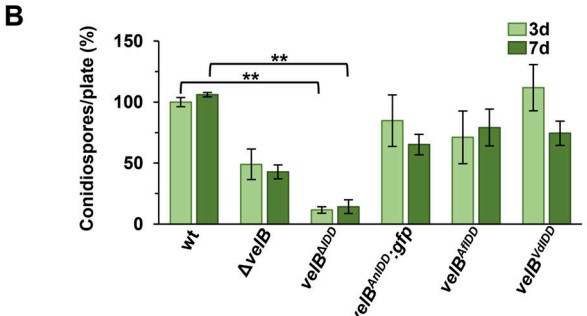

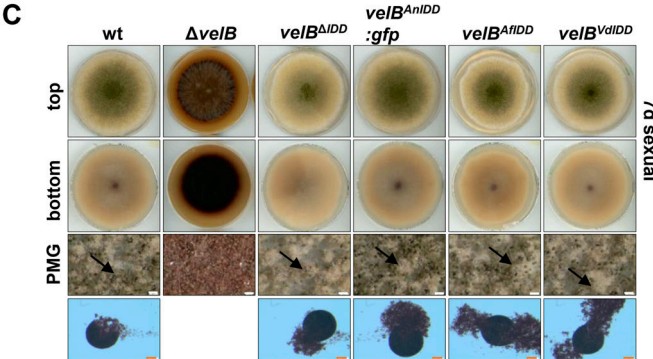

**Figure 4. VelB IDD is required for efficient asexual spore formation in *A. nidulans*.**

**(A)** Phenotypes of WT, *velB* deletion (Δ*velB*), *velB^IDD* deletion (*velB^ΔIDD*), and complementation strains with the IDD of *A. nidulans* (*velB^AnIDD:gfp*), *A. fumigatus* (*velB^AfIDD*), or *Verticillium dahlia* (35) (*velB^VdIDD*) on solid minimal medium. 10^5 spores were distributed over the plate and grown in the light for 3 or 7 d at 37°C. Red arrows indicate aerial hyphae and blue arrows conidiophores. Scale bar = 100 μm. **(A, B)** Quantification of conidiospores from strains shown in (A). Error bars indicate the SEM of three biological replicates. *P*-value was calculated with SD. ***P < 0.005. **(C)** Phenotypes of WT, *velB* deletion, *velB^IDD* deletion, and complementation with the IDD of *A. nidulans*, *A. fumigatus*, or *V. dahliae* into *velB^ΔIDD* strains. Strains were point-inoculated with 10^5 spores on solid minimal medium and incubated in the dark for 7 d at 37°C. Black arrows indicate mature cleistothecia. PMG, photomicrograph; scale bar (white) = 100 μm, (red) = 50 μm.

Quantification of conidiospores after growth with light revealed significantly reduced amounts (12% after 3 d, 14% for the *vel2^ΔIDD* strain after 7 d) relative to WT (Fig 4B). This is similar to the *vel2* deletion strain, which also shows a decrease in conidiospore production. This phenotype was complemented by *in locus* reintroduction of the functional *A. nidulans velB:gfp* fusion construct (*velB^AnIDD:gfp*) or by orthologous sequences from the ascomycete *Aspergillus fumigatus* or *V. dahliae* (*velB^AfIDD*, *velB^VdIDD*). The complementation experiment did not result in any phenotypic change compared with WT. This does not support any strong impact of it the fused GFP domain has on VelB protein stability.

This corroborates that IDD exchanges with different fungi result in functional proteins and can support *A. nidulans* VelB in providing appropriate asexual development and spore formation.

Dehydroaustinol, product of the *aus* gene cluster-encoded proteins, is one of two compounds that signal the induction of sporulation of *A. nidulans* (18, 36). The impact of VelB on the *aus* cluster genes was examined in more detail. The *aus* cluster comprises 14 genes including the four genes *ausI, ausJ, ausM*, and *ausN*, which are essential for austinol and dehydroaustinol biosynthesis (37). They were chosen to test their expression via qRT–PCR in the Δ*velB* and *velB^ΔIDD* strains. Transcript levels of the *ausI, ausJ, ausM*, and *ausN* genes were significantly reduced in the *velB* deletion strain, suggesting that they are controlled by the encoded transcription factor (Fig S10B). Transcription of the *ausJ, ausM*, and *ausN* is significantly reduced in the *velB^ΔIDD* strain, implying an IDD-dependent function in regulation. This suggests that the IDD-dependent VelB-VosA complex is required to activate several genes involved in the biosynthesis of austinol and dehydroaustinol, which stimulates asexual development. These findings are consistent with the decreased conidiospore amounts in the *velB* deletion and the *velB^ΔIDD* mutant strains (Fig 4B).

*A. nidulans* reproduces sexually through the formation of cleistothecia harboring sexual ascospores as overwintering structures in the soil. The *velB* deletion mutant is completely inhibited in sexual reproduction in the dark (Fig 4C). In contrast, the *velB^ΔIDD* mutant strain develops mature cleistothecia filled with ascospores after 7 d in the dark. Therefore, the IDD is not required for sexual development and can also be exchanged by orthologous IDD sequences from *A. fumigatus* or *V. dahliae* without impact on the sexual development.

Whereas VelB IDD and, therefore, the heterodimer VelB-VosA are dispensable for sexual development, the VelB-VeA heterodimer, which still can be formed without the VelB IDD, is absolutely necessary for *A. nidulans* asexual conidiation. The IDD reduces VelB-VeA formation and promotes VelB-VosA formation for induction of the asexual developmental program linked to the appropriate corresponding secondary metabolism.

## The IDD-dependent VelB-VosA heterodimer controls sterigmatocystin production, whereas VelB-VeA controls secondary metabolites relevant for sexual development

Velvet domain proteins connect fungal development with secondary metabolism (18). A *velB* deletion strain secretes dark red-brown pigments into the agar plate, whereas the *velB^ΔIDD* strain shows a lighter color on the bottom of the plate compared with WT

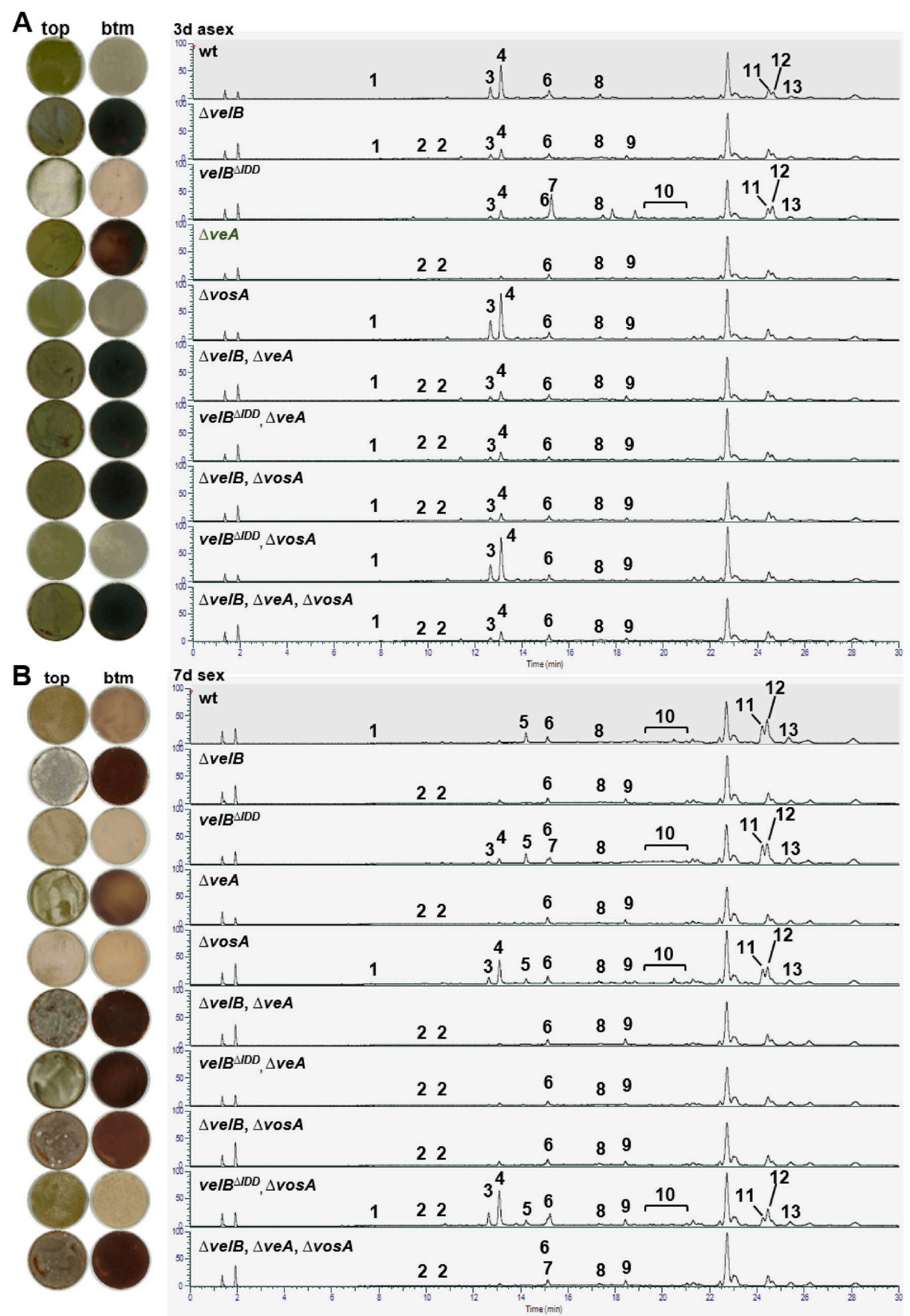

**Figure 5. VelB, VeA, and VosA regulate secondary metabolite production, including VelB IDD control of sterigmatocystin biosynthesis.**
**(A, B)** Charged aerosol detector (CAD) chromatogram of secondary metabolite extracts obtained from 3-d asexual (A) and 7-d sexual (B) incubated strains. The numbers highlight peaks with identified metabolites listed in Table S6: cichorine (**1**, (38)), F-9775A/B (**2**, (39, 40)), austinol (**3**, (37)), dehydroaustinol (**4**, (37)), arugosin H

(Figs 4A and C and 5). This reflects differences in the regulation of secondary metabolism (SM) controlled by the interplay of VelB with its interaction partners. SM analyses were conducted for *velB*, *velB^{ΔIDD}*, *veA*, and *vosA* single, double, and triple deletion strains cultivated for 3 d under asexual or 7 d under sexual development–inducing conditions. The *veA* deletion strain has a similar coloring phenotype like strains without *velB*, but Δ*vosA* shows no red-brown–colored medium and hyphae. SM analysis revealed accumulation of the *orsellinic* (*ors*) cluster product F9775A/B (**2**) in all strains with red-brown color (Figs 5, S11, and S13, Table S6). Loss of *veA* or *vosA* and additional loss of the IDD have no significant impact on the colony color, because double mutants *velB^{ΔIDD}*/Δ*veA* and *velB^{ΔIDD}*/Δ*vosA* resemble the corresponding *veA* and *vosA* single deletion phenotypes, respectively (Fig 5).

SM extracts of Δ*velB*, *velB^{ΔIDD}*, Δ*veA*, and all double deletions strains including Δ*velB* or Δ*veA* display less austinol (**3**) and dehydroaustinol (**4**) compared with the WT or the complementation strain, with the Δ*vosA* strains as the only exceptions (Figs S10A and S11). These data combined with *aus* gene expression experiments underline the importance of VelB for asexual spore formation.

Sterigmatocystin is usually found with low abundance in laboratory WT/reference strains. The *velB^{ΔIDD}* strain produces high amounts of this mycotoxin during asexual development (**7**, Figs 5A, S11A, and S12). LC-MS analysis of SM extracts from sexual development revealed additional sterigmatocystin production in the Δ*vosA* or *velB^{ΔIDD}*/Δ*vosA* strain (Figs 5B and S12). Thin-layer chromatography (TLC) visualized sterigmatocystin (**7**) after derivatization of the compounds on the silica plates with AlCl$_3$. A fivefold increased sterigmatocystin abundance was detected at 366 nm in the *velB^{ΔIDD}* mutant strain after 3 and 7 d of growth in the light compared with WT or *velB^{AnIDD}*:*gfp* complementation strains (Fig S11B and C). Therefore, the VelB-VosA heterodimer possibly acts as a repressor for sterigmatocystin biosynthesis. The *velB^{ΔIDD}*, Δ*vosA*, and *velB^{ΔIDD}*/Δ*vosA* strains show WT-like production of the anthraquinones arugosin H (**5**), arugosin A (**10**), and the xanthones emericellin (**11**), shamixanthone (**12**), and epishamixanthone (**13**) in sexual developmental samples (Fig 5B). These metabolites were not detected in asexual samples, except in the *velB^{ΔIDD}* strain. The VelB-VeA, but not the VelB-VosA, complex influences the production of anthraquinone and xanthone of the *monodictophenone* (*mdp*) cluster, which are important metabolites for sexual development (26).

The LC-MS analysis in addition showed that cichorine (**1**) was increased in Δ*vosA* and *velB^{ΔIDD}*/Δ*vosA*, which is low or absent in all other strains. This suggests that either the VelB-VeA heterodimer has an activating function or the VelB-VosA heterodimer has an inhibitory function on the cichorine cluster. Emericellamide production (**6** and **8**) was decreased in all strains compared with WT under asexual but increased under sexual development–inducing conditions. VelB-VosA positively controls terrequinone A (**9**) synthesis, because it was found in *veA* or *vosA* and especially in the

*velB* deletion strains but was absent in the *velB^{ΔIDD}* strain. This analysis of differences in secondary metabolite formation highlights the importance of precise and accurate control of velvet heterodimer formation to fulfill the distinct molecular functions of the various velvet heterodimer complexes. The IDD allows that VelB can operate specifically depending on its interaction partner as a promoting or as an inhibiting regulator for the formation of appropriate secondary metabolites that are tightly connected with the distinct fungal differentiation programs.

## Discussion

Intrinsically disordered regions within regulators of gene expression contribute to transcriptional activation or DNA binding. Here, we show that the acquisition of an additional intrinsically disordered domain within the DNA-binding and dimerization domain of a single member of a gene family of transcriptional regulators, which can form homo- and heterodimers, allows selective heterodimer formation adjusted to different developmental eukaryotic differentiation programs. A single member of the conserved fungal velvet regulatory gene family allows selective formation of VelB heterodimers either with the velvet domain protein VeA or with VosA. This enables VelB to operate depending on its interaction partner as a positive or negative regulator for a specific differentiation program. VelB-dependent differentiation is connected with the production of appropriate secondary metabolites to communicate with the environment. The ratio of heterodimers of VelB without the IDD is significantly changed toward increased VelB-VeA and strongly decreased VelB-VosA heterodimers (Fig 6). In vitro VosA can form a heterodimer with VelB independently of the IDD. VelB-VosA heterodimer formation without IDD is also possible in vivo in a *veA* deletion strain. More VelB proteins are accessible for VosA binding without the competition of VeA. This suggests a higher VelB-VeA binding affinity than VelB-VosA. It can be assumed that VelB IDD switches into a conformation that restricts VosA interaction under specific conditions during the fungal life cycle.

Fungal velvet and mammalian Rel homology domain (RHD) share a common structure and might even have a common evolutionary ancestor (6, 50). The genetic architecture of these gene families presumably results from gene duplication events with subsequent codon changes for subfunctionalization of original genes (51). Thus, additional functions or interactions to other proteins or DNA can be developed. Mutations in promoters or signal sequences can further alter temporal or spatial protein concentration levels within fungal cells through modulated gene expression (52). DNA acquisition for an intrinsically disordered domain is only found within the fungal *velB* gene family, but neither in other velvet genes nor in the RHD domain family.

---

(**5**, (41, 42)), emericellamide C (**6**, (43, 44, 45)), sterigmatocystin (**7**, (46)), emericellamide E (**8**, (43)), terrequinone A (**9**, (47)), arugosin A (**10**, (41, 48)), emericellin (**11**, (49)), shamixanthone (**12**, (49)), epishamixanthone (**13**, (49)). Structures are shown in Fig S13.

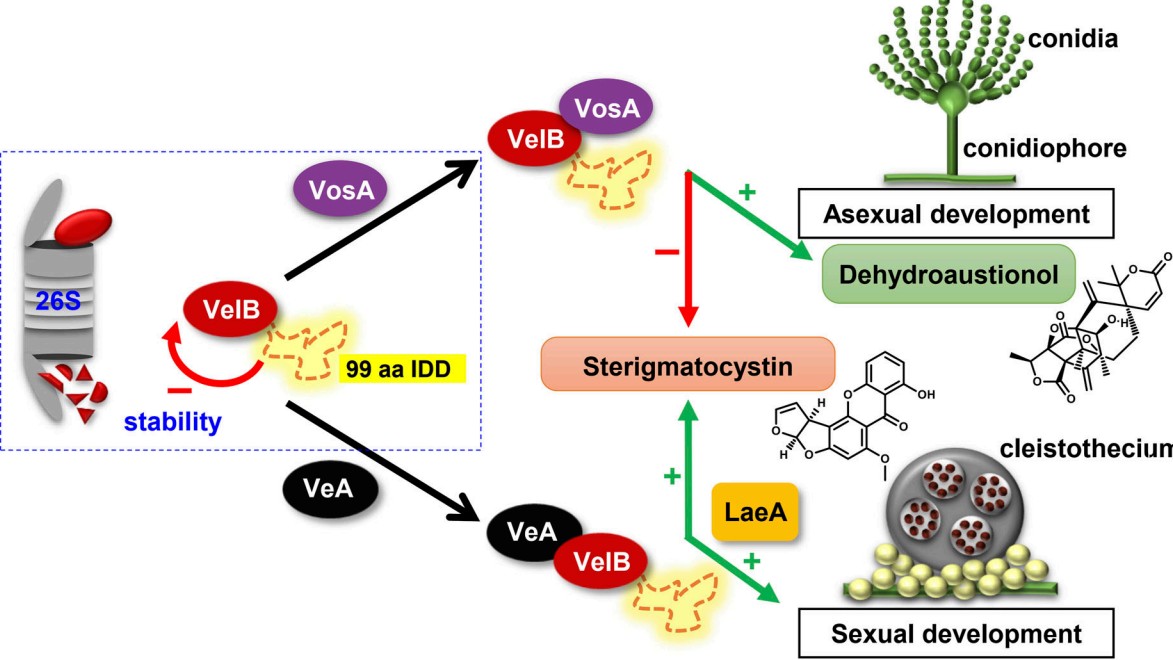

**Figure 6. Model of selective VelB IDD heterodimer formation in *A. nidulans*.**
The VelB dimerization control in *A. nidulans* includes protein stability control and selective heterodimerization. The IDD destabilizes VelB for 26S proteasome degradation. VelB IDD controls selective molecular heterodimer formation. VelB-VosA activates dehydroaustinol production as signal for asexual conidium formation. VelB-VosA reduces sterigmatocystin production either by direct repression or indirectly by reducing VelB levels by competitive VelB-VeA heterodimer formation. Sexual development and sterigmatocystin biosynthesis are induced by VelB-VeA-LaeA.

Higher differentiated Asco- or Basidiomycota mostly contain only one VelB ortholog with IDD. In contrast, members from the phyla Chytridiomycota and Zygomycota have more than one VelB-like isoform where one of them is predicted to be (nearly) continuous, whereas the other isoform mostly has a short interruption of the velvet domain. The VelB copy without IDD might have been lost during evolution. The insertion site of the *velB* DNA for the IDD is conserved during evolution. This supports that the original integration event happened in a common fungal ancestor and was further evolved to the current high diversity of IDD lengths and sequences. Accordingly, chimeric VelB proteins with the IDD region of *A. fumigatus* or *V. dahliae* are functional. The presence of an IDD in the VelB protein is presumably more important than the protein sequence. The VelB IDD involvement in VosA heterodimer formation is conserved between *A. nidulans* and *V. dahliae*, although the Vel2-Vos1 function might differ in the plant pathogen (13).

*A. nidulans* VelB-VeA supports sexual development but delays asexual sporulation as default mode of VelB heterodimer formation for fungal differentiation. Asexual development is accelerated by illumination, which is promoted by VelB-VosA heterodimer through increased dehydroaustinol synthesis (19, 53, 54). Conidiospore numbers or production of the sporulation induction signal dehydroaustinol (36) is significantly reduced in strains without VelB IDD. Consistently, the phenotype of this strain is reminiscent of the appearance of deletion strains impaired in several genes for upstream developmental regulators of conidiation (24, 55). The IDD-dependent VelB-VosA heterodimer further promotes spore survival, whereas deletion of the IDD in *velB* results in reduced spore survival, as described for spores of *velB* or *vosA* deletion strains (56 *Preprint*). The heterodimer activates the expression of the gene *vadA* (VosA/VelB-activated developmental gene), a key regulator of genes such as *brlA*, *rodA*, and *orlA* during sporulation. Deletion of *vadA* resulted in an increased production of cleistothecia during asexual development. In addition, VadA regulation is required for proper amount of trehalose and β(1,3)-glucan, necessary for viability and tolerance to oxidative stress (57). The IDD-dependent VelB-VosA heterodimer also triggers biosynthesis of secondary metabolites including xanthones from the *mdp* gene cluster (26, 49, 58) or the phytotoxin cichorine produced by the nonreducing polyketide synthase CicF (38, 59, 60, 61). VelB-VosA targets the *mcrA* gene encoding another global regulator for secondary metabolism. Accordingly, the *velB^ΔIDD* and a *mcrA* deletion strain share the reduced sporulation phenotype, loss of spore viability, and increased sterigmatocystin production (62).

More than 10% of annually harvested crops are spoiled by fungi and their bioactive metabolites, with a prominent role of the cancerogenic aflatoxin family (63, 64). The *velB^ΔIDD* strain forms fivefold increased amounts of the aflatoxin family compound sterigmatocystin compared with WT. VelB-VeA together with methyltransferase LaeA is activators for sterigmatocystin biosynthesis (18), whereas VelB-VosA reduces this metabolite. Formations of VelB-VeA and presumably VelB-VeA-LaeA are enhanced in the absence of the VelB IDD, whereas VelB-VosA heterodimer formation is nearly

abolished, which elevates sterigmatocystin levels (Fig 6). This is consistent with the increased expression of biosynthetic genes and sterigmatocystin production in ascospores of a *vosA* deletion strain (65). The comparative analysis of *A. nidulans* WT with different combinations of velvet deletion strains revealed that each strain produced a different secondary metabolite peak profile (Fig 5). This suggests that the IDD fine-tunes fungal secondary metabolism and provides a valid tool to dissect the impact of the different VelB heterodimers on fungal secondary metabolite control.

The acquisition of the VelB IDD provides a protein-destabilizing function, in addition to the control of heterodimer partner selection. Half-lives of fungal transcription factors controlling multiple genes in different fungal differentiation or pathogenicity programs vary considerably (66, 67, 68). The interplay between the expression of the *velB* gene and IDD-mediated protein stability determines the cellular VelB levels and adjusts fungal development and secondary metabolism. The ubiquitin-mediated 26S proteasomal degradation system reduces half-lives of various proteins with internal disordered domains of more than 40 residues (69, 70). The different VelB IDDs of the fungal kingdom share numerous serine residues (Fig S2). These are potential sites of phosphorylation as priming reaction for subsequent ubiquitination and degradation (71, 72). NetPhos 3.1 predicts multiple putative phosphorylation sites between amino acids 133 and 149 in the IDD of VelB (73, 74). This region falls within a consistent gap in peptide coverage from residues 131 to 236, confirmed both experimentally and bioinformatically (Expasy PeptideCutter) (75), which prevents direct detection of potential PTMs. Posttranslational modifications and masking or demasking of the IDD could provide molecular mechanism to increase or decrease VelB stability and shift the ratio of VelB protein complexes in response to different environmental stimuli in *A. nidulans* (Fig 6). Stability of the VelB interaction partner VeA depends on a complex interplay between the destructing ubiquitinating F-box23 containing E3 cullin-RING ligase for labeling for proteasomal degradation and the reversal stabilization by the deubiquitinating enzyme UspA (29, 76). The VelB IDD presumably interferes with numerous protein interactions, because interaction partners with different cellular functions were exclusively identified in pull-downs of VelB^ΔIDD-GFP but not of VelB-GFP. Many of these carry a conventional nuclear localization signal (34). The IDD could provide VelB conformational flexibility and structural plasticity as a variable hub, which allows the interaction with different proteins during various environmental conditions. These interactions with other proteins can also reduce or increase the affinity toward VosA. Different interaction partners control or adjust cellular locations of distinct VelB complexes within or outside of the nucleus. This is illustrated by the finding that a VelB protein without IDD and therefore unable to interact to VosA is unable to enter the nucleus, when the VeA protein is also missing in the cell.

The IDD-mediated proportion of cellular VelB heterodimers therefore includes two control levels: (i) the IDD selects between VeA and VosA and (ii) the IDD stabilizes or destabilizes VelB as a heterodimer-binding partner. This allows to respond to different environmental stimuli in favor of specific fungal developmental programs linked to the appropriate secondary metabolism. These external stimuli that initiate either asexual or sexual developmental programs are, for example, light or darkness. They lead to a preferred VeA-VelB heterodimer during sexual and a more predominant heterodimer VelB-VosA during asexual development.

Here, we discovered a novel mechanism for an intrinsically disordered domain within a velvet transcription factor, which has been specifically introduced and further evolved within the VelB protein family in the fungal kingdom. Velvet proteins control fungal defense, including the entire genetic network of fungal development, virulence, and secondary metabolism, whereas the mammalian NF-κB proteins with a similar DNA-binding fold are relevant for infection or immune defense. The VelB IDD coordinates and fine-tunes the fungal chemical language by controlling the ratio of VelB heterodimer formation and their stability. This might be a promising starting point for a better understanding of fungal communication. In addition, it will be interesting to examine whether there will be IDDs with similar functions for heterodimeric transcription factors in other organisms than the fungi.

# Materials and Methods

### Strains and growth conditions

*A. nidulans* strains were cultivated in liquid or solid minimal medium (MM) (77) in the light with oxygen supply–inducing asexual development or in darkness with limited oxygen supply by sealing the plates with parafilm-inducing sexual development. For details, see reference 24. *V. dahliae* strains were cultivated as described in reference 13.

### Plasmid and strain preparation

All strains used in this study are listed in Table S7. A list of plasmids used in this study is shown in Table S8, and oligonucleotides in Table S9. Genomic DNA of FGSC A4 (*A. nidulans* WT, *veA*^+ (78)) was used as a template for amplification of DNA fragments for plasmid constructions. Gene targeting by homologous recombination was performed with recyclable marker (RM) cassettes (79). Amplified DNA fragments and recyclable marker cassettes, which were excised from pME4304 and pME4305 with *Sfi*I, and are called natRM and phleoRM, respectively, were cloned into the *EcoR*V multiple cloning site of pBluescript SK(+) using a seamless cloning reaction (Invitrogen). Transformation of plasmid excised cassettes into *A. nidulans* was performed by polyethylene glycol-mediated protoplast fusion as described earlier (80). *V. dahliae* was transformed as described in reference 81. Transformation of plasmids into *E. coli* was conducted as described before (82, 83). *E. coli* strains were cultivated in lysogeny broth (LB) (84) medium (1% [wt/vol] tryptophan, 0.5% [wt/vol] yeast extract, 1% [wt/vol] NaCl).

### Plasmid and strain construction of *velB*^ΔIDD

For construction of a *velB*^ΔIDD strain 1, the 5′ flanking region and half of the velvet domain of the *velB* gene (to exclude the IDD) were amplified with SR110/SR109. The second half of the *velB* velvet domain was

amplified with SR108/SR111. The 3′ flanking region was amplified with SR112/SR113. These fragments and the phleoRM cassette were cloned to pBluescript SK(*) resulting in pME4686. The velB$^{ΔIDD}$ cassette was excised with PmeI and transformed to AGB551 (85) resulting in AGB1131 into AGB1066 giving AGB1140 and AGB1057 giving AGB1142.

## Plasmid and strain construction of velB:gfp

gfp was amplified with primers SR18/SR20 from pME4292. The 5′ flanking region and the velB gene were amplified with SR05/SR24. The 3′ flanking region was amplified with SR07/SR08. These three fragments and the natRM marker cassette were cloned into pBluescript SK(+), resulting in pME4687. The velB:gfp cassette was excised with PmeI, followed by transformation in AGB551 and AGB1066, which resulted in the AGB1132 and AGB1190, respectively.

## Plasmid and strain construction of velB$^{ΔIDD}$:gfp

Fragments from constructing pME4686 were used for the assembly of this construct, but the second half of the velB velvet domain was amplified with primers SR108/SR24. The velB velvet domain fragments amplified with SR109/SR110 and SR108/SR24 were joined by fusion PCR. This desired fragment and gfp (amplified with SR18/SR20 from pME4292), the 3′ flanking region, and the phleoRM marker cassette were cloned into pBluescript SK(+) giving plasmid pME4688. The velB$^{ΔIDD}$:gfp cassette was excised with PmeI, followed by transformation in AGB551 and AGB1066, which resulted in AGB1133 and AGB1191, respectively.

## Plasmid and strain construction of velB$^{AfIDD}$ complementation

The 3′ flanking region was amplified with SR112/SR113 and cloned into the Eco72I restriction site of pME4319 (86), resulting in pME4689. The 5′ flanking region and half of the velvet domain of the A. nidulans velB gene (until the IDD) were amplified with primers SR110/SR253. The 0.3 kb IDD of A. fumigatus velB was amplified with SR266/SR267 from A. fumigatus Afs35 gDNA. The second half of the A. nidulans velB velvet domain was amplified with SR108/SR111. All fragments were joined by fusion PCR. The fused fragment was cloned into the SwaI restriction site of pME4689, resulting in pME4690. The velB$^{AfIDD}$ complementation cassette was excised with PmeI, followed by transformation in AGB1133, resulting in AGB1134.

## Plasmid and strain construction of velB$^{VdIDD}$ complementation

The 0.4-kb IDD of V. dahliae vel2 was amplified with SR254/SR255 from V. dahliae JR2 gDNA. The fragments SR110/253, SR254/255, and SR108/111 were joined by fusion PCR. The fused fragment was cloned into the SwaI restriction site of pME4689, resulting in pME4691. The velB$^{VdIDD}$ complementation cassette was excised with PmeI, followed by transformation in AGB1133, yielding AGB1135.

## Plasmid construction of pME4692 for recombinant expression of velB$^{ΔIDD}$:his in E. coli

The fragments for the two parts of the velB velvet domain were amplified from pME3815 with primers JG45/SR109 and JG46/SR108. These fragments were joined by fusion PCR containing an overhang for the NcoI or XhoI restriction site, respectively. After subcloning in the pJET plasmid, the construct was excised with NcoI and XhoI and cloned into the NcoI/XhoI site of pETM-13, which contains a sequence encoding a C-terminal His-tag, resulting in pME4692. The plasmid was transformed to Rosetta II E. coli strain (NOVAGEN, MERCK) for recombinant protein expression.

## Plasmid and strain construction of vosA:ha

The 5′ flanking region and the vosA gene were amplified with primers SR76/SR201, introducing a sequence encoding a HA (hemagglutinin antigen)-tag. The 3′ flanking region was amplified with primers SR49/SR75. The two fragments and the phleoRM marker cassette were cloned into the EcoRV multiple cloning site of pBluescript SK(+), resulting in pME4693. The vosA:ha cassette was excised with PmeI and transformed into AGB1132 and AGB1133, resulting in AGB1136 and AGB1137, respectively.

## Plasmid and strain construction of veA:ha

The veA 5′ flanking region was amplified with veA using KT197/KT166. This template was used for another PCR introducing a sequencing encoding a HA-tag using KT197/KT163. The seamless cloning kit was used to ligate the 5′:veA:ha fragment with the natRM and the veA 3′ flanking region (KT142/KT198) into pBluescript SK(+) resulting in pME4748. The veA:ha cassette was excised with PmeI and transformed into AGB1132 and AGB1133, resulting in AGB1149 and AGB1150, respectively.

## Plasmid and strain construction of V. dahliae vel2$^{ΔIDD}$:GFP

The construction of a VEL2 strain without IDD fused to GFP was conducted in several steps. In the first step, the primers AO74 and AO75 were used to amplify the 5′ flanking region of the gene and VEL2 until the start of the IDD from gDNA. In another PCR, AO76 and AO77 were used to amplify the part of VEL2 downstream of the IDD from gDNA. As AO75 was constructed with an overhang to VEL2 after the IDD, the two fragments were fused by PCR using AO74 and AO77. The fragment was ligated in the EcoRV-linearized pPK2 (87), and the resulting plasmid was named pME5075. In the next step, pME5075 was cut with XbaI and the 3′ flanking region of VEL2 was amplified with AO78 and AO79. Ligation of pME5075 and the PCR fragment resulted in a plasmid named pME5076. In the last step, the 5′ flanking region and VEL2 without IDD and stop codon were amplified with AO167 and AO168 from pME5076. The 3′ flanking region was amplified with the primers AO169 and AO170. GFP (without start codon) and a flexible linker (protein sequence GGSGG) were amplified from pME4990 with AO165 and RH514. The hygromycin resistance cassette was amplified from the same plasmid with the primers RH590 and RO4. The GFP-linker fragment and the hygromycin marker were fused by PCR with the primers

AO165 and RO4. The three generated fragments were ligated into pME4564 (88 Preprint) and cut with EcoRV and StuI. The created plasmid was named pME5077. The WT (JR2) was transformed with the plasmid resulting in VGB468. The gDNA of the constructed strain was treated with HincII and PstI and tested by Southern hybridization with the 5′ flanking region as a probe. The strain was also confirmed by cutting the gDNA with EcoRI and conducting a Southern hybridization with the 3′ flanking region as a probe.

## Extraction of fungal genomic DNA

Extraction of A. nidulans genomic DNA from liquid cultures was performed as described before (24). Therefore, A. nidulans strains were inoculated in 300-ml flasks containing 100 ml MM by shaking overnight at 37°C to generate mycelia. The mycelia were harvested and frozen in liquid nitrogen, followed by pulverization. Genomic DNA of V. dahliae was extracted as described in reference 89.

## Isolation of fungal RNA and cDNA synthesis and quantitative real-time PCR

Fungal RNA was isolated from vegetative cultures. Subsequent cDNA synthesis and quantitative real-time PCR were performed as described before (24). cDNA synthesis for checking the expression of velB was performed with 0.8 $\mu$g RNA and for checking the expression of aus genes with 2.0 $\mu$g RNA. Primers used for qRT–PCRs are listed in Table S10.

## Sterigmatocystin isolation

$1 \times 10^5$ spores were distributed on solid MM and grown for 3 or 7 d at 37°C in the light or dark. Two agar plugs were excised with a 50-ml centrifuge tube (Sarstedt), which were cut into small pieces. The agar pieces were transferred into 50-ml Falcon tubes, and six glass beads and 3 ml $H_2O$ were added. Samples were shaken roughly for 30 min at RT. Subsequently, 3 ml chloroform was added and samples were incubated for another 30 min at RT. This was followed by centrifugation at 1,000 rpm for 10 min at 4°C for phase separation, and the lower chloroform metabolite-containing chloroform phase was transferred into glass tubes and evaporated o/n at RT under the hood.

## Thin-layer chromatography

Samples of sterigmatocystin isolation were resuspended in 50 $\mu$l methanol, and three times 5 $\mu$l of isolated sterigmatocystin per sample was applied spot-wise to precoated SIL G/UV254 Polygram DC-foil TLC sheets (Macherey-Nagel) (thin-layer chromatography plates). TLC plates were run in 1:4 (vol/vol) acetone:chloroform for 40–50 min, dried for 5 min, and photographed at 366 and 254 nm with a Camag TLC Visualizer 2 system (Camag). Afterward, TLC plates were sprayed with 20% (vol/vol) aluminum chloride in 95% (vol/vol) ethanol and incubated at 70°C for 10 min. Derivatized TLC plates were photographed again at 366- and 254-nm UV light and white light with a Camag TLC visualizer 2 system and processed with winCATS 1.4.4 software (Camag).

## Secondary metabolite extraction

For secondary metabolite extraction, $1 \times 10^5$ spores were distributed over the whole agar plate and grown for 3 d at 37°C in the light, promoting asexual development. Subsequently, four round agar pieces with a diameter of 2.5 cm were excised and homogenized with a syringe. Metabolites were extracted with a mixture of 8 ml water and 8 ml ethyl acetate overnight. After centrifugation, 5 ml of the organic phase was dried, dissolved in 500 $\mu$l methanol, and subjected to LC-MS analysis.

## LC-MS analysis of secondary metabolites

The reconstituted metabolites were analyzed using a Q Exactive Focus Orbitrap mass spectrometer coupled with an UltiMate 3000 HPLC (Thermo Fisher Scientific). 1 $\mu$l of each sample was injected on a HPLC column (Acclaim 120, C18, 5 $\mu$m, 120 Å, 4.6 × 100 mm [Thermo Fisher Scientific]) applying a linear acetonitrile/0.1% (vol/vol) formic acid in $H_2O$/0.1% (vol/vol) formic acid gradient (from 5% to 95% [vol/vol] acetonitrile/0.1 formic acid in 20 min, plus additional 10 min with 95% [vol/vol] acetonitrile/0.1 formic acid) with a flow rate of 0.8 ml/min at 30°C. The measurements were performed in a mass range of 70-1,050 m/z in positive or negative mode. Data analysis was performed with Thermo Scientific Xcalibur 4.1 (Thermo Fisher Scientific) and FreeStyle 1.4 (Thermo Fisher Scientific). The identified secondary metabolites in this study are listed in Table S6.

## Phenotypic analyses of fungal strains

For phenotypic analyses, either 2,000 spores were spotted in the middle of the agar plate or $10^5$ spores were distributed over the whole plate. Agar plates were incubated for 3 or 7 d under either asexual or sexual development–promoting conditions. Photomicrographic pictures of A. nidulans colonies were obtained by the use of an Axiolab microscope (Zeiss) or with the help of a binocular microscope SZX12-ILLB2-200 (Olympus). Visualization was performed with an Olympus SC30 digital camera, and pictures were processed with cellSens software (Olympus).

## Fluorescence microscopy

Fluorescence microscopy was conducted with a Zeiss AxioObserver Z.1 inverted confocal microscope, equipped with Plan-Neofluar 63x/0.75 (air), Plan-Apochromat 63x/1.4 oil, and Plan-Apochromat 100x/1.4 oil objectives (Zeiss), and a QuantEM:512SC camera (Photometrics). Pictures were processed with the SlideBook 6.0 software package (Intelligent Imaging Innovations).

For fluorescence microscopy, 2000 spores per strain were inoculated in eight-well borosilicate cover glass system (Thermo Fisher Scientific) in 400 $\mu$l liquid MM or on agar slants for vegetative growth and grown for 18 h at 30°C. Nuclei were visualized via staining with 0.1% (wt/vol) DAPI (Roth) and incubated for 10 min at RT before microscopy.

### Conidiospore quantification

Conidiospore numbers were determined by the use of a Coulter Z2 particle counter (Beckman coulter) or with a Thoma cell counting chamber (hemocytometer) (Paul Marienfeld).

### Protein isolation

$1 \times 10^7$ spores were inoculated in MM, and strains were grown vegetatively in submerged cultures. For protein isolation of asexually or sexually grown cultures, strains were grown vegetatively for 24 h and subsequently shifted onto solid MM plates and grown for indicated time points in the light or dark. For the cycloheximide assay, after 24 h of vegetative growth, 250 µl cycloheximide (10 mg/ml) was added to the cultures, which were incubated for 0–5 h before protein extraction. Mycelia were collected, and protein crude extracts were obtained as described previously (24).

### Immunoprecipitation with GFP-tagged fusion proteins

$5 \times 10^8$ spores of *A. nidulans* strains were inoculated in 500 ml MM and grown for 24 h in submerged cultures (vegetative samples) at 37°C. Sexual samples were shifted afterward to solid MM, sealed with Parafilm· and incubated for 48 h in the dark at 37°C. Protein GFP pull-downs were conducted by employing GFP-Trap_A beads from (Chromotek) as described earlier (59, 66).

### Pull-downs with HA-tagged fusion proteins

Protein HA pull-downs were conducted using Monoclonal Anti-HA-Agarose beads A2095 (Sigma-Aldrich). *A. nidulans* strains were inoculated in a concentration of $5 \times 10^8$ spores in 500 ml MM and vegetatively grown for 24 h in submerged cultures at 37°C. Mycelia were treated like for GFP trap. HA beads were washed with PBS, equilibrated with B⁺ buffer, added to the filtered supernatant, and incubated with rotating for 3 h or o/n at 4°C. The supernatant with the HA beads was loaded onto fresh Poly-Prep Chromatography Columns (Bio-Rad), which were equilibrated with B⁺ buffer before, and washed with W500 buffer. Proteins were eluted with 100 µl 0.1 M glycine, pH 2.5, and 2.5 µl Tris, pH 10.4, and analyzed by Western experiments.

### SDS–PAGE and Western experiments

SDS–polyacrylamide gel electrophoresis (SDS–PAGE) and Western experiments were performed as described in reference 66. Primary antibodies anti-GFP (sc-9996; SANTA CRUZ BIOTECHNOLOGY), anti-tubulin (T0926; Sigma-Aldrich), 6xHis-tag monoclonal antibody (R930-25; Thermo Fisher Scientific), anti-GST (Z-5; SANTA CRUZ BIOTECHNOLOGY), and anti-hemagglutinin (anti-HA, clone HA-7; Sigma-Aldrich) were diluted in TBST-M (TBST buffer, supplemented with 5% [wt/vol] skim milk powder) and incubated o/n at 4°C. Secondary antibodies, either horseradish peroxidase–coupled rabbit antibody (G21234; INVITROGEN) or mouse antibody (115-035-003; JACKSON IMMUNORESEARCH), were used in a dilution of 1:1,000 in TBST-M. As a loading control, membranes were stained with Ponceau staining. The GFP and HA signals and the Ponceau signals were analyzed with Bio1D software (version 15.08; Vilber Lourmat Deutschland GmbH). For normalization of the Western experiment data, the Ponceau signal was used. The calculation of the significance was done using an unpaired $t$ test for two groups using mean, SEM, and number of samples.

### In vitro co-immunoprecipitation of recombinant expressed proteins

For in vitro co-IPs, LB was inoculated with a preculture of the respective *E. coli* cells and incubated on a rotary shaker. Protein expression was induced with 1 mM IPTG at $OD_{600}$ of 0.8. After o/n incubation at 20°C on a rotary shaker, cells were harvested by centrifugation at 1912.3g for 20 min at 4°C, washed with buffer W (100 mM Tris, pH 8, 150 mM NaCl, 1 mM EDTA), and centrifuged again. For protein purification and in vitro co-IP, the cell pellets were resuspended in buffer W supplemented with 1 mM PMSF and cell lysis was performed by sonication. VosA-GST (90) cells were centrifuged at 50,288g for 30 min at 4°C. The supernatant was incubated with GST beads for 3 h with rotating at 4°C. After incubation, the lysate was centrifuged at 119.5g for 2 min at 4°C and washed with buffer W + 1 mM IPTG. The beads were divided into two 15-ml reaction tubes, and the supernatant of lysed cells containing either pME3815 or pME4692 was added and incubated for 2 h on a rotator at 4°C followed by centrifugation at 119.5g for 2 min at 4°C. The beads were washed twice with buffer W1 mM IPTG, transferred to 1.5-ml reaction tubes, and washed twice again. The supernatant was removed carefully from the beads, and 3x SDS sample buffer was added and boiled for 10 min at 95°C. The samples were analyzed by 12% SDS–PAGE, and experiments for specific detection of His- or GST-tagged proteins.

### Tryptic protein digestion and peptide analysis with LC-MS

Protein samples were separated by SDS–PAGE. The gel was incubated in fixing solution (40% [vol/vol] ethanol, 10% [vol/vol] acetic acid) for 1 h, washed with $dH_2O$, and incubated o/n in colloidal Coomassie (91) (5% [wt/vol] aluminum-sulfate-(14-18)-hydrate, 10% [vol/vol] methanol, 0.1 [wt/vol] Coomassie Brilliant Blue G-250, 2% [vol/vol] orthophosphoric acid). Each lane was excised completely and cut into small pieces of ~2 mm. Protein tryptic digestion was performed as described earlier using Sequencing Grade Modified Trypsin (Promega) (24, 92), followed by StageTip purification as described in reference 93, 94.

### Identification of proteins by LC-MS/MS

Peptide solutions were analyzed with mass spectrometry coupled to liquid chromatography (LC-MS) at the LC-MS facility at the Institute of Microbiology and Genetics as described in reference 13, 29, 59. The samples were subjected to reverse phase liquid chromatography for peptide separation using an RSLCnano Ultimate 3000 system (Thermo Fisher Scientific). Peptides were loaded on an Acclaim PepMap 100 precolumn (100 µm × 2 cm, C18, 5 µm, 100 Å; Thermo Fisher Scientific) with 0.07% trifluoroacetic acid at a flow rate of 20 µl/min for 3 min. Analytical separation of peptides

was done on an Acclaim PepMap RSLC column (75 µm × 50 cm, C18, 2 µm, 100 Å; Thermo Fisher Scientific) at a flow rate of 300 nl/min. The solvent composition was gradually changed within 94 min from 96% solvent A (0.1% formic acid) and 4% solvent B (80% acetonitrile, 0.1% formic acid) to 10% solvent B within 2 min, to 30% solvent B within the next 58 min, to 45% solvent B within the following 22 min, and to 90% solvent B within the last 12 min of the gradient. Eluting peptides were online ionized by nano-electrospray (nESI) using the Nanospray Flex Ion Source (Thermo Fisher Scientific) and transferred into an Orbitrap mass spectrometer (Thermo Fisher Scientific). Full scans in a mass range of 300 to 1,650 m/z were recorded at a resolution of 30,000 followed by data-dependent top 10 HCD fragmentation at a resolution of 15,000 (dynamic exclusion enabled). LC-MS method programming and data acquisition were performed with Xcalibur 4.0 software (Thermo Fisher Scientific). Protein database searches and subsequent data analyses were carried out using MaxQuant (V.2.2.0.0) and Perseus (V.2.0.7.0), respectively.

## Data Availability

All other data are found in the article and supplemental material.

## Supplementary Information

## Acknowledgements

The authors thank Verena Große, Nicole Scheiter, Helen Stupperich, and Sarah E Eubanks for their technical assistance. We would also like to thank Prof. Dr. Ralf Ficner and Dr. Piotr Neumann for providing the structure and the resulting figure. We acknowledge support by the Open Access Publication Funds of the University of Göttingen. This work was supported by funding from the German Research Foundation (DFG: https://www.dfg.de) to GH Braus (BR1502/15-2, BR1502/18-2, and IRTG PRoTECT). LC-MS for metabolite analysis was funded by the Deutsche Forschungsgemeinschaft (INST 186/1287-1 FUGG). The funders had no role in study design, data collection and analysis, decision to publish, or preparation of the article.

### Author Contributions

AM Köhler: conceptualization, formal analysis, validation, investigation, visualization, methodology, and writing—original draft, review, and editing.
S Thieme: conceptualization, formal analysis, validation, investigation, visualization, methodology, and writing—original draft, review, and editing.
J Gerke: conceptualization, supervision, validation, investigation, and writing—review and editing.
KG Thieme: formal analysis, validation, investigation, visualization, and writing—review and editing.
R Harting: conceptualization, supervision, validation, visualization, and writing—original draft, review, and editing.
K Schmitt: formal analysis, validation, investigation, and writing—review and editing.
O Valerius: formal analysis, validation, investigation, and writing—review and editing.
W Chen: investigation, visualization, and writing—review and editing.
A Höfer: formal analysis, validation, investigation, visualization, and writing—review and editing.
E Bastakis: investigation, visualization, and writing—review and editing.
A Strohdiek: visualization and writing—review and editing.
E-S Xylakis: validation and writing—review and editing.
AK Heinrich: investigation and writing—review and editing.
HB Bode: resources, investigation, and writing—review and editing.
GH Braus: conceptualization, resources, supervision, funding acquisition, and writing—original draft, review, and editing.

### Conflict of Interest Statement

The authors declare that they have no conflict of interest.

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
