## [Reviewer comments · Life Science Alliance]

The VelB IDD promotes selective heterodimer formation of velvet proteins for fungal development

Anna Köhler, Sabine Thieme, Jennifer Gerke, Karl Thieme, Rebekka Harting, Kerstin Schmitt, Oliver Valerius, Wanping Chen, Annalena Höfer, Emmanouil Bastakis, Anja Strohdiek, Emmanouil-Stavros Xylakis, Antje Heinrich, Helge Bode, and Gerhard Braus

DOI: <https://doi.org/10.26508/lsa.202503395>

Corresponding author(s): Gerhard Braus, University of Göttingen

Review Timeline:

Submission Date:	2025-05-21
Editorial Decision:	2025-07-07
Revision Received:	2025-08-20
Editorial Decision:	2025-09-25
Revision Received:	2025-10-29
Accepted:	2025-10-31

Scientific Editor: Tim Fessenden

Transaction Report:

July 7, 2025

Re: Life Science Alliance manuscript #LSA-2025-03395-T

Prof. Gerhard H Braus
Georg-August-University
Institute of Microbiology & Genetics
Molecular Microbiology and Genetics
Grisebachstr. 8
Göttingen D-37077
Germany

Dear Dr. Braus,

Thank you for submitting your manuscript entitled "The VelB IDD promotes selective heterodimer formation of velvet proteins for fungal development" to Life Science Alliance. The manuscript was assessed by expert reviewers, whose comments are appended to this letter.

As you will see, both reviewers appreciate the intriguing new findings on Vel protein interactions that modify their regulation of gene expression in fungi. However Reviewer 1 noted that suitable loading controls for co-IP assays were missing from Figure 3. Reviewer 3 requested improved evidence to confirm that the IDD of VelB modulates the stability of this protein. In addition to these additions which are required in a revision, both reviewers made several suggestions to improve this work which we invite you to consider.

Thank you for this interesting contribution to Life Science Alliance. We are looking forward to receiving your revised manuscript.

Sincerely,

-- Summary blurb (enter in submission system): A short text summarizing in a single sentence the study (max. 200 characters including spaces). This text is used in conjunction with the titles of papers, hence should be informative and complementary to

the title and running title. It should describe the context and significance of the findings for a general readership; it should be written in the present tense and refer to the work in the third person. Author names should not be mentioned.

B. MANUSCRIPT ORGANIZATION AND FORMATTING:

Reviewer #1 (Comments to the Authors (Required)):

In their manuscript, Köhler et al. Describe the detailed functional analysis of an intrinsically disordered domain (IDD) that is present within the velvet domain of fungal *velB* proteins. Analysis of Velvet proteins from across the fungal kingdom showed that the IDD within the velvet domain is restricted to *VelB* proteins. Functional analysis of the IDD in the *VelB* protein of the ascomycete *Aspergillus nidulans* showed that presence of the IDD promotes protein degradation and is required for interaction with the *VosA* protein in vivo, but not in vitro. The latter finding can be explained by a preferential binding of *VelB* to *VeA*, supported by in vivo *VosA* binding even without the IDD in a *veA* deletion background. In the absence of *veA*, the IDD also promotes nuclear localization of *VelB*, most likely because nuclear localization requires interaction with *VosA*, which is weaker in the absence of the IDD. Phenotypic analyses showed that the IDD is required for wild type-like conidiation, but not for sexual development. Interestingly, exchange of the *A. nidulans* IDD with the IDDs of *Aspergillus fumigatus* or *Verticillium dahliae* did not interfere with its function in development, suggesting a conserved evolutionary function of the IDD despite limited sequence conservation. Analysis of secondary metabolites revealed that sterigmatocystin production was strongly increased in the absence of the IDD, and overall metabolite profiles were distinct in strains with different combinations of mutants of members of the velvet family. The results highlight the role of intrinsically disordered regions in regulating protein interactions and thereby fine-tuning the effects of transcription factors in different developmental stages and/or environmental conditions, a topic has not yet been explored in great depths in fungi.

There are just a few points where the manuscript could be improved:

1. Lines 396-397: it is not clear why the domain PDB 2MLY was chosen for modeling the *VelB* IDD apart from a similar size. It seems unlikely that a random domain of similar size would give a useful structure for modeling. Please give more explanation.
2. Lines 420-423 and Fig. S2: In Fig. S2, the chytrids are actually shown in part C and the zygomycota in part D (not the other way round as stated in the Figure legend). The conserved region in the velvet domain that flanks the C-terminal end of the IDD seems to be partly conserved not only in basidiomycetes, but also in zygomycota, where the motif ends with TRN in many species (Fig. S2D). Thus it appears to be conserved in fungi except for chytrids and it might be good to include this in Figure 1 for the basidiomycetes and zygomycota.
3. Lines 429-431 and Table S6: In Table S6 in the zygomycota, I don't see any species with an IDD length of zero, rather there are many with short IDDs of 6 amino acids. It might be interesting to check if these 6 amino acids are conserved.
4. Lines 443-445: Please describe the strains in more detail. Are these strains where the wild type copy of *velB* was substituted by the versions fused to *gfp* and/or without the region encoding the IDD or are these strains where a *delta-velB* was complemented (ectopically) with the corresponding constructs? This should be clarified throughout the manuscript.
5. Figure 2: In the Western blots for the *VelB-deltaIDD*, it looks as if there is always a double band (in Figures 2A and 2B). Is there an explanation for that (e.g. from the mass spectrometry, were there different variants of the protein detected)?
6. Lines 477-479, Fig. S4: The transcript levels don't really look similar, rather it looks as if the transcript level for *velB* is upregulated after 18 h whereas for *velB-deltaIDD* this is not the case.
7. Line 538: Please define the abbreviation Lfq.

8. Figure 3: In Figure 3A, the descriptions of what is shown in the different images is missing. However, it might be better to combine Figures 3A and 3D into one Figure so that readers can better compare the situation in the wild type with that in the analyzed mutant strains. For Figures 3B and 3C (and described in lines 574-576), it is not clear what the bar graphs refer to. Are these the relative signal strengths in the IP? For the IP experiments, also the Western blots for the input (before the IP) should be shown to compare relative input amounts with amounts in the IP, especially for the VosA experiments where no interaction with the VelB-deltaIDD was detected (e.g. to exclude that the stability of VelB with or without IDD influences the stability of VosA).

9. Figure 4: The image quality of the shown developmental stages is not very good. If this is not just a problem with the pdf conversion, images with higher resolution should be provided.

10. Supplementary Tables: There seems to be a mismatch with the numbering of the Supplementary Tables in the actual tables and the text (starting with Table S1 in line 121, which appears to refer to the actual Table S7 in the supplementary material). This should be checked throughout.

11. In Table S9 (oligonucleotides), please indicate which oligonucleotides were used for amplification of which genes/genomic regions.

12. Figure S6: Legend for Figure S6D is missing.

Reviewer #3 (Comments to the Authors (Required)):

Review of the manuscript LSA-2025-03395-T

The velvet regulators govern development and secondary metabolism in the filamentous fungus *Aspergillus nidulans*. The regulator VelB carries an intrinsically disordered domain (IDD), which is conserved within the fungal kingdom. IDDs of transcription factors often contribute to DNA binding specificity and protein interactions. The aim of the authors was to elucidate the regulatory function of VelB's IDD and its impact on the interaction with the velvet complex protein VeA and the velvet regulator VosA. This was achieved by the generation of a velB IDD knock-out strain, various velB, veA deletion strains, the construction of a VelB-GFP fusion protein expressing strain, GFP-pull down experiments, recombinant production of VelB, as well as secondary metabolite analyses. The authors showed that the IDD destabilises VelB and enables the heterodimer formation with VosA to promote asexual development. In contrast, heterodimer formation of VelB with VeA is favoured over VosA and does not need the presence of the IDD. In addition, the IDD-dependent formation of the VelB-VosA heterodimer regulates sterigmatocystin production, while the VelB-VeA heterodimer controls secondary metabolites produced during sexual development.

Major points

The manuscript addresses a fundamental and highly relevant research topic in the field of fungal biology and is based on solid experiment. The manuscript provides more further insights in the complex regulation of the velvet transcription factors in fungi. There are a few points I recommend to be considered:

1) It still remains obscure, by which mechanism the IDD increases the heterodimer formation of VelB with VosA and how the IDD influences the protein stability. Which signals or intracellular modifiers alter the IDD conformation in such a way that VosA or VeA preferentially bind to VelB? Do the authors have any hint from their own data or from the literature (including phosphoproteomics) that the IDD of VelB is modified by PTMs? Do the authors have a hypothesis how the IDD has an impact on the affinity to VosA? Do other proteins have an impact on this interaction?

2) The authors state that the IDD influences protein stability. However, the experimental data on the influence of IDD on protein stability are not convincing. The analysis of protein stability as a function of the IDD is based on only one method, Western blot analysis using anti-GFP.

Are the different VelB levels also detectable in targeted proteomics analysis?

Can the authors exclude that the GFP fusion itself has an influence on the stability?

The additional bands (approx. 55 and 40 kDa) in VelB-GFP are not discussed.

Also, the quantification (Fig. 2B) using Ponceau staining with faint signals is not very reliable.

3) Fig. 2: It is not explained why no full-length fusion protein is detectable after 18h but free GFP is still present (line 482). Why does the degradation of the VelB-GFP protein stop at GFP?

4) The description of some experiments does not provide the number of technical or biological replicates that were performed. For example, how many replicated were performed for the pull-down experiments listed in table 1. Do the values represent mean values?

5) Can the authors exclude that the IDD has also an effect on the stability of the VelB mRNA transcript?

6) It cannot be concluded from Fig. S4 that the expression of *velB*-gfp and *velB* IDD-gfp is "similar" (line 478). After 18h, for example, there is clearly more *velB*-gfp transcript than *velB* IDD-gfp transcript. Which statistical analysis was carried out to verify the significance?

Minor points

The arrangement of the figures in Figure 3 does not correspond to the legend. Figure 3B does not show the localization of *VelB*-GFP in *velA*, *laeA* and *vosA* deletion strains, as described in the legend, but the pull-down data, which are shown in 3D according to the legend. This must be corrected.

In several places, "wildtype" must be corrected to "wild type".

p. 10, line 268

The authors should add a sentence about the identification of the secondary metabolites and in addition they could refer to Table S6.

p. 12, line 327

How were the Western blot signals detected and quantified?

p. 33, line 831

The authors could also shortly mention and discuss the regulator *VadA* in this context.

p. 21, Table 1

MS/MS counts is a less common term and should be explained or replaced by "Peptide Spectrum Matches". Or do the authors mean something different?

Dear Editors,

We have addressed all comments and suggestions as follows:

- A letter addressing the reviewers' comments point by point.
 - We added the answers to the reviewer comments below.
- An editable version of the final text (.DOC or .DOCX) is needed for copyediting (no PDFs).
 - All files are available as word documents.
- High-resolution figure, supplementary figure and video files uploaded as individual files: See our detailed guidelines for preparing your production-ready images.
 - We uploaded all figures and supplementary figures as individual files. The legends for the supplementary figures were therefore added at the end of the manuscript. The legends of the main figures were left after the paragraph where they were first mentioned.
- Summary blurb (enter in submission system): A short text summarizing in a single sentence the study (max. 200 characters including spaces). This text is used in conjunction with the titles of papers, hence should be informative and complementary to the title and running title. It should describe the context and significance of the findings for a general readership; it should be written in the present tense and refer to the work in the third person. Author names should not be mentioned.
 - We added the following sentence into the submission system: Selective velvet domain heterodimer formation promotes fungal development
- Full guidelines are available on our Instructions for Authors page, <https://www.life-science-alliance.org/authors>
 - We changed the formatting style of the manuscript following the guidelines. Therefore, we would like to point out to the reviewers that their comments containing line numbers no longer correspond to the current line numbers.

We have an additional comment: We added one co-author (E. S. Xylakis), who has provided the proteomics analysis for the revised version of the manuscript. All co-authors agreed to his co-authorship.

Reviewer #1 (Comments to the Authors (Required)):

In their manuscript, Köhler et al. describe the detailed functional analysis of an intrinsically disordered domain (IDD) that is present within the velvet domain of fungal velB proteins. Analysis of Velvet proteins from across the fungal kingdom showed that the IDD within the velvet domain is restricted to VelB proteins. Functional analysis of the IDD in the VelB protein of the ascomycete *Aspergillus nidulans* showed that presence of the IDD promotes protein degradation and is required for interaction with the VosA protein in vivo, but not in vitro. The latter finding can be explained by a preferential binding of VelB to VeA, supported by in vivo VosA binding even without the IDD in a veA deletion background. In the absence of veA, the IDD also promotes nuclear localization of VelB, most likely because nuclear localization requires interaction with VosA, which is weaker in the absence of the IDD. Phenotypic analyses showed that the IDD is required for wild type-like conidiation, but not for sexual development. Interestingly, exchange of the *A. nidulans* IDD with the IDDs of *Aspergillus fumigatus* or *Verticillium dahliae* did not interfere with its function in development, suggesting a conserved evolutionary function of the IDD despite limited sequence conservation. Analysis of secondary metabolites revealed that sterigmatocystin production was strongly increased in the absence of the IDD, and overall metabolite profiles were distinct in strains with different combinations of mutants of members of the velvet family. The results highlight the role of intrinsically disordered regions in regulating protein interactions and thereby fine-tuning the effects of transcription factors in different developmental stages and/or environmental conditions, a topic has not yet been explored in great depths in fungi.

There are just a few points where the manuscript could be improved:

1. Lines 396-397: it is not clear why the domain PDB 2MLY was chosen for modeling the VelB IDD apart from a similar size. It seems unlikely that a random domain of similar size would give a useful structure for modeling. Please give more explanation. This 99 amino acids VelB-IDD domain was removed during crystallization for the Xray analysis by the routine protease treatment of the crystals and is therefore not visible in the Xray structure. The PDB 2MLY domain has a similar size as VelB-IDD and was therefore chosen for modeling the VelB IDD part (line 130-135).

2. Lines 420-423 and Fig. S2: In Fig. S2, the chytrids are actually shown in part C and the zygomycota in part D (not the other way round as stated in the Figure legend). The conserved region in the velvet domain that flanks the C-terminal end of the IDD seems to be partly conserved not only in basidiomycetes, but also in zygomycota, where the motif end with TRN in many species (Fig. S2D). Thus it appears to be conserved in fungi except for chytrids and it might be good to include this in Figure 1 for the basidiomycetes and zygomycota.

This TRN motif is indeed conserved in fungi except chytrids, but is located outside of the IDD domain as part of the conserved velvet domain. We have addressed this point to avoid misunderstandings and have rearranged Figure 1B (line 157-160, Fig 1). In addition, the (C) Chytridiomycota and (D) Zygomycota are now exchanged (Legend Fig 2S, line 1185-1186).

3. Lines 429-431 and Table S6: In Table S6 in the zygomycota, I don't see any species with an IDD length of zero, rather there are many with short IDDs of 6 amino acids. It might be interesting to check if these 6 amino acids are conserved.

We have changed the text according to the suggestion: The Zygomycetes carry genes for VelB with very short IDDs. Whereas the lack of the IDD (zero amino acids was never observed, there are short Zygomycetes IDDs with six amino acids, which very not conserved in their sequence in other fungi (Table S5 and S6)" (line 166-169).

4. Lines 443-445: Please describe the strains in more detail. Are these strains where the wild type copy of *velB* was substituted by the versions fused to *gfp* and/or without the region encoding the IDD or are these strains where a delta-*velB* was complemented (ectopically) with the corresponding constructs? This should be clarified throughout the manuscript.

We added that both strains were created by complementing the $\Delta velB$ strain within the locus. Line 184-185.

5. Figure 2: In the Western blots for the VelB-deltaIDD, it looks as if there is always a double band (in Figures 2A and 2B). Is there an explanation for that (e.g. from the mass spectrometry, were there different variants of the protein detected)?

The VelB^{ΔIDD}-GFP fusion protein showed a double band compared to the full length VelB-GFP, suggesting different posttranslational modifications (PTMs) of VelB^{ΔIDD}-GFP (lines 234-236). The mass spectrometry indicated a phosphorylation site at threonine 84 (T84) specific to the VelB^{ΔIDD}. The different post-translational modification states of VelB^{ΔIDD} might result from the phosphorylation at T84. We have added this point in the text (lines 236-239)

6. Lines 477-479, Fig. S4: The transcript levels don't really look similar, rather it looks as if the transcript level for *velB* is upregulated after 18 h whereas for *velB*-deltaIDD this is not the case.

The relative normalized expression of *velB*^{ΔIDD} compared to *velB* was similar from six to 6h incubation under asexual conditions (Fig S4). The transcript level of *velB* was higher compared to *velB*^{ΔIDD} after 18h of asexual development. The constant level of *velB*^{ΔIDD} transcripts resembles the protein amount of VelB^{ΔIDD}-GFP (Fig 2B, Fig S4, line 1193, line 1196), (line 215-220).

7. Line 538: Please define the abbreviation LFQ.

The definition of the abbreviation was added to line 297.

8. Figure 3: In Figure 3A, the descriptions of what is shown in the different images is missing. However, it might be better to combine Figures 3A and 3D into one Figure so that readers can better compare the situation in the wild type with that in the analyzed mutant strains. For Figures 3B and 3C (and described in lines 574-576), it is not clear what the bar graphs refer to. Are these the relative signal strengths in the IP? For the IP experiments, also the Western blots for the input (before the IP) should be shown to compare relative input amounts with amounts in the IP, especially for the VosA experiments where no interaction with the VelB-deltaIDD was detected (e.g. to exclude that the stability of VelB with or without IDD influences the stability of VosA).

Figure A and D were now placed into one figure. Therefore, text parts were rearranged from the end of the paragraph to line 251-253 and 259-264.

We added more information in the text to clarify what is shown in the graphs (line 329-331). The graphs show the relative protein amount of the VeIB-GFP and VeIB^{ΔIDD}-GFP which was compared regarding the protein amount of the corresponding VeA-HA or VosA-HA proteins.

9. Figure 4: The image quality of the shown developmental stages is not very good. If this is not just a problem with the pdf conversion, images with higher resolution should be provided.

We have set the pixel density significantly higher to improve the quality.

10. Supplementary Tables: There seems to be a mismatch with the numbering of the Supplementary Tables in the actual tables and the text (starting with Table S1 in line 121, which appears to refer to the actual Table S7 in the supplementary material). This should be checked throughout.

This was checked and corrected through the document.

11. In Table S9 (oligonucleotides), please indicate which oligonucleotides were used for amplification of which genes/genomic regions.

We placed now in the table S9 which regions were amplified by the primers.

12. Figure S6: Legend for Figure S6D is missing.

The legend for Fig S6D was added under the figure (line 1234-1235).

Reviewer #3 (Comments to the Authors (Required)):

Review of the manuscript LSA-2025-03395-T: The velvet regulators govern development and secondary metabolism in the filamentous fungus *Aspergillus nidulans*. The regulator VelB carries an intrinsically disordered domain (IDD), which is conserved within the fungal kingdom. IDDs of transcription factors often contribute to DNA binding specificity and protein interactions. The aim of the authors was to elucidate the regulatory function of VelB's IDD and its impact on the interaction with the velvet complex protein VeA and the velvet regulator VosA. This was achieved by the generation of a *velB*ΔIDD knock-out strain, various *velB*, *veA* deletion strains, the construction of a VelB-GFP fusion protein expressing strain, GFP-pull down experiments, recombinant production of VelB, as well as secondary metabolite analyses. The authors showed that the IDD destabilises VelB and enables the heterodimer formation with VosA to promote asexual development. In contrast, heterodimer formation of VelB with VeA is favoured over VosA and does not need the presence of the IDD. In addition, the IDD-dependent formation of the VelB-VosA heterodimer regulates sterigmatocystin production, while the VelB-VeA heterodimer controls secondary metabolites produced during sexual development.

Major points

The manuscript addresses a fundamental and highly relevant research topic in the field of fungal biology and is based on solid experiment. The manuscript provides more further insights in the complex regulation of the velvet transcription factors in fungi. There are a few points I recommend to be considered:

1) It still remains obscure, by which mechanism the IDD increases the heterodimer formation of VelB with VosA and how the IDD influences the protein stability.

- Which signals or intracellular modifiers alter the IDD conformation in such a way that VosA or VeA preferentially bind to VelB?
 - A suggestion to this question was added to line 647-650, where we state that the VeA-VelB heterodimer is preferred during sexual and a more predominant heterodimer VelB-VosA during asexual development.
- Do the authors have any hint from their own data or from the literature (including phosphoproteomics) that the IDD of VelB is modified by PTMs?
 - NetPhos 3.1 predicts multiple putative phosphorylation sites between amino acids 133 and 149 in the IDD of VelB (Beide Blom Paper). This region falls within a consistent gap in peptide coverage from residues 131 to 236, confirmed both experimentally and bioinformatically (ExPASy PeptideCutter), which prevents direct detection of potential PTMs. The information was added to line 622-626.
- Do the authors have a hypothesis how the IDD has an impact on the affinity to VosA?
- Do other proteins have an impact on this interaction?
 - For both questions, we added one more sentence in the section about VelB/Protein interactions (line 638-639) to clarify that the IDD plays a role in the affinity towards VosA as well as other proteins, which interact with VelB, can have an impact on this interaction.

2) The authors state that the IDD influences protein stability. However, the experimental data on the influence of IDD on protein stability are not convincing. The analysis of protein stability as a function of the IDD is based on only one method, Western blot analysis using anti-GFP.

- Are the different VelB levels also detectable in targeted proteomics analysis?
 - VelB was detectable by label-free quantification, and TSIM was performed for peptides outside the IDD region. However, due to limited coverage within the IDD, these data cannot fully resolve IDD-dependent stability differences. Targeted analysis of the IDD region would be needed for confirmation. This was added in the result part in line 338-342.
- Can the authors exclude that the GFP fusion itself has an influence on the stability?
 - The complementation experiment did not result in any phenotypical change compared with wild type. This does not support any strong impact of it the fused GFP domain has on VelB protein stability (line 416-419).
- The additional bands (approx. 55 and 40 kDa) in VelB-GFP are not discussed. Also, the quantification (Fig. 2B) using Ponceau staining with faint signals is not very reliable.
 - The additional ~55 and ~40 kDa bands in the VelB-GFP western experiment likely represent truncated or degraded forms of the protein. Mass spectrometry detected internal VelB peptides for the 55 kDa band but lacked N- or C-terminal sequences, supporting this. (Line 228-231)

3) **Fig. 2:** It is not explained why no full-length fusion protein is detectable after 18h but free GFP is still present (line 482). Why does the degradation of the VelB-GFP protein stop at GFP?

The degradation of VelB-GFP and VelB^{ΔIDD}-GFP does not affect the resulting free GFP. GFP has a stable protein structure and no GFP-specific protein degradation pathways exist, therefore its degradation occurs slowly (line 231-234).

4) The description of some experiments does not provide the number of technical or biological replicates that were performed. For example, how many replicated were performed for the pull-down experiments listed in table 1. Do the values represent mean values?

The values for LFQ intensity, MS/MS counts, and Unique peptides represent the mean value from three biological replicates (Line 307-311). The conducted replicates were also added to the analysis of the sterigmatocystin analysis Fig S11 (line 1281). We also added this to Fig S5 (line 1207-1208).

5) Can the authors exclude that the IDD has also an effect on the stability of the VelB mRNA transcript?

The mRNA did not show a significant difference for the transcript levels of VelB and VelB^{ΔIDD} IDD, which does not support that there is a significant IDD effect on the VelB mRNA transcript (Line 215-220).

6) It cannot be concluded from **Fig. S4** that the expression of velB-gfp and velB^{ΔIDD}-gfp is "similar" (line 478). After 18h, for example, there is clearly more velB-gfp

transcript than *velB* Δ IDD-gfp transcript. Which statistical analysis was carried out to verify the significance?

We corrected that the relative normalized expression of *velB* Δ IDD compared to *velB* was similar from six to 6h incubation under asexual conditions. The transcript level of *velB* was higher compared to *velB* Δ IDD after 18h of asexual development. Since the experiment was conducted with two biological replicates and three technical replicates, no statistical test for significance was performed. This experiment is only intended to indicate that the difference in the western experiment is not based on the level of transcription (line 215-220).

Minor points

The arrangement of the figures in **Figure 3** does not correspond to the legend. Figure 3B does not show the localization of VelB-GFP in *velA*, *laeA* and *vosA* deletion strains, as described in the legend, but the pull-down data, which are shown in 3D according to the legend. This must be corrected.

We corrected the figure, so that it the legend fits to the figure.

In several places, "**wildtype**" must be corrected to "wild type".

Wildtype was changed to wild type throughout the text.

p. 10, line 268: The authors should add a sentence about the identification of the secondary metabolites and in addition they could refer to Table S6.

In the method section about secondary metabolites was now added that the secondary metabolites identified in this study are listed in table S10 (Line 815-816).

p. 12, line 327: How were the Western blot signals detected and quantified?

Information about the detection of the signals from the western experiments and their quantification was added to line 875-880.

p. 33, line 831: The authors could also shortly mention and discuss the regulator VadA in this context.

We stated in line 585-590 the role of the regulator VadA in the context of sporulation and its activation by the VelB-VosA heterodimer.

p. 21, Table 1: MS/MS counts is a less common term and should be explained or replaced by "Peptide Spectrum Matches". Or do the authors mean something different?

MS/MS count is now explained under the table in line 310-311 that is refers to the number of specific peptides fragmented and analyzed.

Hopefully, all points were covered adequate.

We are looking forward to hearing from you.

Sincerely yours,

Gerhard Braus

September 25, 2025

RE: Life Science Alliance Manuscript #LSA-2025-03395-TR

Prof. Gerhard H Braus
University of Göttingen
Institute of Microbiology & Genetics
Molecular Microbiology and Genetics
Grisebachstr. 8
Göttingen D-37077
Germany

Dear Dr. Braus,

Thank you for submitting your revised manuscript entitled "The VelB IDD promotes selective heterodimer formation of velvet proteins for fungal development". We returned this to the reviewers, however only Reviewer 1 was available to evaluate this revision. We would be happy to publish your paper in Life Science Alliance pending final revisions necessary to meet our formatting guidelines.

- Please include references from the supporting information file in the main reference list.
- Please add the X and Bluesky handles of your host institute/organization, as well as your own and/or one of the authors, in our system.
- Please move your main, supplementary figure, and table legends to the main manuscript text after the references section.
- Tables should be included at the bottom of the main manuscript file or sent as separate files.
- Please include a Data Availability section after the Materials & Methods section. Please consider providing an accession number to the mass spec data at a public repository. Please consult our guidelines at <https://www.life-science-alliance.org/manuscript-prep#format>
- Please add an Author Contributions section to your main manuscript text.
- Please add a Conflict of Interest statement to your main manuscript text.
- We encourage you to revise the figure legend for Figure 3 such that the figure panels are introduced in alphabetical order.
- There are callouts for figure S12A-B, and this figure doesn't have these panels...please correct.
- Please add callouts for Figures 3D; S2A-D; S5D; S5D and S7 to your main manuscript text.
- Please provide full details for genomic DNA extraction, and please add details on mycelia collection and crude protein extraction, in the Materials and Methods section.

A. FINAL FILES:

- An editable version of the final text (.DOC or .DOCX) is needed for copyediting (no PDFs).
- High-resolution figure, supplementary figure and video files uploaded as individual files: See our detailed guidelines for preparing your production-ready images, <https://www.life-science-alliance.org/authors>
- Summary blurb (enter in submission system): A short text summarizing in a single sentence the study (max. 200 characters

including spaces). This text is used in conjunction with the titles of papers, hence should be informative and complementary to the title. It should describe the context and significance of the findings for a general readership; it should be written in the present tense and refer to the work in the third person. Author names should not be mentioned.

B. MANUSCRIPT ORGANIZATION AND FORMATTING:

Thank you for your attention to these final processing requirements. Please revise and format the manuscript and upload materials as soon as you are able.

Sincerely,

Reviewer #1 (Comments to the Authors (Required)):

The authors have addressed the concerns that I had about the first version of the manuscript.

Dear Editors,

We have addressed all comments and suggestions as follows:

- Please include references from the supporting information file in the main reference list.
 - We added all the references also to the main reference list.
- Please move your main, supplementary figure, and table legends to the main manuscript text after the references section.
 - We added all the legends under the reference section.
- Tables should be included at the bottom of the main manuscript file or sent as separate files.
 - We sent the table as a separate file.
- Please include a Data Availability section after the Materials & Methods section.
 - We included this section after Material and Methods.
- Please add an Author Contributions section to your main manuscript text.
 - Author contribution was added under the main manuscript.
- Please add a Conflict of Interest statement to your main manuscript text.
 - We added the statement to the main manuscript.
- We encourage you to revise the figure legend for Figure 3 such that the figure panels are introduced in alphabetical order.
 - We changed the alphabetical order of the figure panels (line 1142, line 1147).
- There are callouts for figure S12A-B, and this figure doesn't have these panels...please correct.
 - The figure S12 has no A and B panel.
- Please add callouts for Figures 3D; S2A-D; S5D; S5D and S7 to your main manuscript text.
 - We added the callouts for S2A-D in line 136, 140, and 143; for S5D in line 293; for S7 in line 306).
 - Figure 3 has no D panel; therefore, it was not added in the manuscript text.
- Please provide full details for genomic DNA extraction, and please add details on mycelia collection and crude protein extraction, in the Materials and Methods section.
 - We added more information for gDNA extraction in line 680-682. Preparation of crude extracts is described in line 747-753.

- A letter addressing the reviewers' comments point by point.
 - We added the answers to the reviewer comments below.
- An editable version of the final text (.DOC or .DOCX) is needed for copyediting (no PDFs).
 - All files are available as word documents.
- High-resolution figure, supplementary figure and video files uploaded as individual files: See our detailed guidelines for preparing your production-ready images.
 - We uploaded all figures and supplementary figures as individual files. The legends for the supplementary figures were therefore added at the end of the manuscript. The legends of the main figures were left after the paragraph where they were first mentioned.
- Summary blurb (enter in submission system): A short text summarizing in a single sentence the study (max. 200 characters including spaces). This text is used in conjunction with the titles of papers, hence should be informative and complementary to the title and running title. It should describe the context and significance of the findings for a general readership; it should be written in the present tense and refer to the work in the third person. Author names should not be mentioned.
 - We added the following sentence into the submission system: Selective velvet domain heterodimer formation promotes fungal development
- Full guidelines are available on our Instructions for Authors page, <https://www.life-science-alliance.org/authors>
 - We changed the formatting style of the manuscript following the guidelines. Therefore, we would like to point out to the reviewers that their comments containing line numbers no longer correspond to the current line numbers.

We have an additional comment: We added one co-author (E. S. Xylakis), who has provided the proteomics analysis for the revised version of the manuscript. All co-authors agreed to his co-authorship.

Reviewer #1 (Comments to the Authors (Required)):

In their manuscript, Köhler et al. describe the detailed functional analysis of an intrinsically disordered domain (IDD) that is present within the velvet domain of fungal VelB proteins. Analysis of Velvet proteins from across the fungal kingdom showed that the IDD within the velvet domain is restricted to VelB proteins. Functional analysis of the IDD in the VelB protein of the ascomycete *Aspergillus nidulans* showed that presence of the IDD promotes protein degradation and is required for interaction with the VosA protein in vivo, but not in vitro. The latter finding can be explained by a preferential binding of VelB to VeA, supported by in vivo VosA binding even without the IDD in a veA deletion background. In the absence of veA, the IDD also promotes nuclear localization of VelB, most likely because nuclear localization requires interaction with VosA, which is weaker in the absence of the IDD. Phenotypic analyses showed that the IDD is required for wild type-like conidiation, but not for sexual development. Interestingly, exchange of the *A. nidulans* IDD with the IDDs of *Aspergillus fumigatus* or *Verticillium dahliae* did not interfere with its function in development, suggesting a conserved evolutionary function of the IDD despite limited sequence conservation. Analysis of secondary metabolites revealed that sterigmatocystin production was strongly increased in the absence of the IDD, and overall metabolite profiles were distinct in strains with different combinations of mutants of members of the velvet family. The results highlight the role of intrinsically disordered regions in regulating protein interactions and thereby fine-tuning the effects of transcription factors in different developmental stages and/or environmental conditions, a topic has not yet been explored in great depths in fungi.

There are just a few points where the manuscript could be improved:

1. Lines 396-397: it is not clear why the domain PDB 2MLY was chosen for modeling the VelB IDD apart from a similar size. It seems unlikely that a random domain of similar size would give a useful structure for modeling. Please give more explanation. **This 99 amino acids VelB-IDD domain was removed during crystallization for the Xray analysis by the routine protease treatment of the crystals and is therefore not visible in the Xray structure. The PDB 2MLY domain has a similar size as VelB-IDD and was therefore chosen for modeling the VelB IDD part (line 1180-1185).**

2. Lines 420-423 and Fig. S2: In Fig. S2, the chytrids are actually shown in part C and the zygomycota in part D (not the other way round as stated in the Figure legend). The conserved region in the velvet domain that flanks the C-terminal end of the IDD seems to be partly conserved not only in basidiomycetes, but also in zygomycota, where the motif end with TRN in many species (Fig. S2D). Thus it appears to be conserved in fungi except for chytrids and it might be good to include this in Figure 1 for the basidiomycetes and zygomycota.

This TRN motif is indeed conserved in fungi except chytrids, but is located outside of the IDD domain as part of the conserved velvet domain. We have addressed this point to avoid misunderstandings and have rearranged Figure 1B (line 140-143, Fig 1). In addition, the (C) Chytridiomycota and (D) Zygomycota are now exchanged (Legend Fig 2S, line 1185-1186).

3. Lines 429-431 and Table S6: In Table S6 in the zygomycota, I don't see any species with an IDD length of zero, rather there are many with short IDD's of 6 amino acids. It might be interesting to check if these 6 amino acids are conserved.

We have changed the text according to the suggestion: The Zygomycetes carry genes for VelB with very short IDD's. Whereas the lack of the IDD (zero amino acids was never observed, there are short Zygomycetes IDD's with six amino acids, which very not conserved in their sequence in other fungi (Table S5 and S6)" (line 149-152).

4. Lines 443-445: Please describe the strains in more detail. Are these strains where the wild type copy of *velB* was substituted by the versions fused to *gfp* and/or without the region encoding the IDD or are these strains where a delta-*velB* was complemented (ectopically) with the corresponding constructs? This should be clarified throughout the manuscript.

We added that both strains were created by complementing the $\Delta velB$ strain within the locus. Line 167-168.

5. Figure 2: In the Western blots for the VelB-deltaIDD, it looks as if there is always a double band (in Figures 2A and 2B). Is there an explanation for that (e.g. from the mass spectrometry, were there different variants of the protein detected)?

The VelB^{ΔIDD}-GFP fusion protein showed a double band compared to the full length VelB-GFP, suggesting different posttranslational modifications (PTMs) of VelB^{ΔIDD}-GFP (lines 198-200). The mass spectrometry indicated a phosphorylation site at threonine 84 (T84) specific to the VelB^{ΔIDD}. The different post-translational modification states of VelB^{ΔIDD} might result from the phosphorylation at T84. We have added this point in the text (lines 200-203).

6. Lines 477-479, Fig. S4: The transcript levels don't really look similar, rather it looks as if the transcript level for *velB* is upregulated after 18 h whereas for *velB*-deltaIDD this is not the case.

The relative normalized expression of *velB*^{ΔIDD} compared to *velB* was similar from six to 6h incubation under asexual conditions (Fig S4). The transcript level of *velB* was higher compared to *velB*^{ΔIDD} after 18h of asexual development. The constant level of *velB*^{ΔIDD} transcripts resembles the protein amount of VelB^{ΔIDD}-GFP (Fig 2B, Fig S4, line 1298, line 1301), (line 179-184).

7. Line 538: Please define the abbreviation LFQ.

The definition of the abbreviation was added to line 237.

8. Figure 3: In Figure 3A, the descriptions of what is shown in the different images is missing. However, it might be better to combine Figures 3A and 3D into one Figure so that readers can better compare the situation in the wild type with that in the analyzed mutant strains. For Figures 3B and 3C (and described in lines 574-576), it is not clear what the bar graphs refer to. Are these the relative signal strengths in the IP? For the IP experiments, also the Western blots for the input (before the IP) should be shown to compare relative input amounts with amounts in the IP, especially for the VosA experiments where no interaction with the VelB-deltaIDD was detected (e.g. to exclude that the stability of VelB with or without IDD influences the stability of VosA).

Figure A and D were now placed into one figure. Therefore, text parts were rearranged from the end of the paragraph to line 223-228 and 260-262.

We added more information in the text to clarify what is shown in the graphs (line 260-262). The graphs show the relative protein amount of the VeIB-GFP and VeIB^{ΔIDD}-GFP which was compared regarding the protein amount of the corresponding VeA-HA or VosA-HA proteins.

9. Figure 4: The image quality of the shown developmental stages is not very good. If this is not just a problem with the pdf conversion, images with higher resolution should be provided.

We have set the pixel density significantly higher to improve the quality.

10. Supplementary Tables: There seems to be a mismatch with the numbering of the Supplementary Tables in the actual tables and the text (starting with Table S1 in line 121, which appears to refer to the actual Table S7 in the supplementary material). This should be checked throughout.

This was checked and corrected through the document.

11. In Table S9 (oligonucleotides), please indicate which oligonucleotides were used for amplification of which genes/genomic regions.

We placed now in the table S9 which regions were amplified by the primers.

12. Figure S6: Legend for Figure S6D is missing.

The legend for Fig S6D was added under the figure (line 1337-1338).

Reviewer #3 (Comments to the Authors (Required)):

Review of the manuscript LSA-2025-03395-T: The velvet regulators govern development and secondary metabolism in the filamentous fungus *Aspergillus nidulans*. The regulator VelB carries an intrinsically disordered domain (IDD), which is conserved within the fungal kingdom. IDDs of transcription factors often contribute to DNA binding specificity and protein interactions. The aim of the authors was to elucidate the regulatory function of VelB's IDD and its impact on the interaction with the velvet complex protein VeA and the velvet regulator VosA. This was achieved by the generation of a *velB*ΔIDD knock-out strain, various *velB*, *veA* deletion strains, the construction of a VelB-GFP fusion protein expressing strain, GFP-pull down experiments, recombinant production of VelB, as well as secondary metabolite analyses. The authors showed that the IDD destabilises VelB and enables the heterodimer formation with VosA to promote asexual development. In contrast, heterodimer formation of VelB with VeA is favoured over VosA and does not need the presence of the IDD. In addition, the IDD-dependent formation of the VelB-VosA heterodimer regulates sterigmatocystin production, while the VelB-VeA heterodimer controls secondary metabolites produced during sexual development.

Major points

The manuscript addresses a fundamental and highly relevant research topic in the field of fungal biology and is based on solid experiment. The manuscript provides more further insights in the complex regulation of the velvet transcription factors in fungi. There are a few points I recommend to be considered:

1) It still remains obscure, by which mechanism the IDD increases the heterodimer formation of VelB with VosA and how the IDD influences the protein stability.

- Which signals or intracellular modifiers alter the IDD conformation in such a way that VosA or VeA preferentially bind to VelB?
 - A suggestion to this question was added to line 548-551, where we state that the VeA-VelB heterodimer is preferred during sexual and a more predominant heterodimer VelB-VosA during asexual development.
- Do the authors have any hint from their own data or from the literature (including phosphoproteomics) that the IDD of VelB is modified by PTMs?
 - NetPhos 3.1 predicts multiple putative phosphorylation sites between amino acids 133 and 149 in the IDD of VelB (Beide Blom Paper). This region falls within a consistent gap in peptide coverage from residues 131 to 236, confirmed both experimentally and bioinformatically (ExPASy PeptideCutter), which prevents direct detection of potential PTMs. The information was added to line 523-527.
- Do the authors have a hypothesis how the IDD has an impact on the affinity to VosA?
- Do other proteins have an impact on this interaction?
 - For both questions, we added one more sentence in the section about VelB/Protein interactions (line 539-540) to clarify that the IDD plays a role in the affinity towards VosA as well as other proteins, which interact with VelB, can have an impact on this interaction.

2) The authors state that the IDD influences protein stability. However, the experimental data on the influence of IDD on protein stability are not convincing. The analysis of protein stability as a function of the IDD is based on only one method, Western blot analysis using anti-GFP.

- Are the different VelB levels also detectable in targeted proteomics analysis?
 - VelB was detectable by label-free quantification, and TSIM was performed for peptides outside the IDD region. However, due to limited coverage within the IDD, these data cannot fully resolve IDD-dependent stability differences. Targeted analysis of the IDD region would be needed for confirmation. This was added in the result part in line 269-273.
- Can the authors exclude that the GFP fusion itself has an influence on the stability?
 - The complementation experiment did not result in any phenotypical change compared with wild type. This does not support any strong impact of it the fused GFP domain has on VelB protein stability (line 347-350).
- The additional bands (approx. 55 and 40 kDa) in VelB-GFP are not discussed. Also, the quantification (Fig. 2B) using Ponceau staining with faint signals is not very reliable.
 - The additional ~55 and ~40 kDa bands in the VelB-GFP western experiment likely represent truncated or degraded forms of the protein. Mass spectrometry detected internal VelB peptides for the 55 kDa band but lacked N- or C-terminal sequences, supporting this. (Line 192-195)

3) **Fig. 2:** It is not explained why no full-length fusion protein is detectable after 18h but free GFP is still present (line 482). Why does the degradation of the VelB-GFP protein stop at GFP?

The degradation of VelB-GFP and VelB^{ΔIDD}-GFP does not affect the resulting free GFP. GFP has a stable protein structure and no GFP-specific protein degradation pathways exist, therefore its degradation occurs slowly (line 195-198).

4) The description of some experiments does not provide the number of technical or biological replicates that were performed. For example, how many replicated were performed for the pull-down experiments listed in table 1. Do the values represent mean values?

The values for LFQ intensity, MS/MS counts, and Unique peptides represent the mean value from three biological replicates (Line 1266-1269). The conducted replicates were also added to the analysis of the sterigmatocystin analysis Fig S11 (line 1376). We also added this to Fig S5 (line 1302-1303).

5) Can the authors exclude that the IDD has also an effect on the stability of the VelB mRNA transcript?

The mRNA did not show a significant difference for the transcript levels of VelB and VelB^{ΔIDD} IDD, which does not support that there is a significant IDD effect on the VelB mRNA transcript (Line 179-184).

6) It cannot be concluded from **Fig. S4** that the expression of velB-gfp and velB^{ΔIDD}-gfp is "similar" (line 478). After 18h, for example, there is clearly more velB-gfp

transcript than *velB* Δ IDD-gfp transcript. Which statistical analysis was carried out to verify the significance?

We corrected that the relative normalized expression of *velB* Δ IDD compared to *velB* was similar from six to 6h incubation under asexual conditions. The transcript level of *velB* was higher compared to *velB* Δ IDD after 18h of asexual development. Since the experiment was conducted with two biological replicates and three technical replicates, no statistical test for significance was performed. This experiment is only intended to indicate that the difference in the western experiment is not based on the level of transcription (line 179-184).

Minor points

The arrangement of the figures in **Figure 3** does not correspond to the legend. Figure 3B does not show the localization of VelB-GFP in *velA*, *laeA* and *vosA* deletion strains, as described in the legend, but the pull-down data, which are shown in 3D according to the legend. This must be corrected.

We corrected the figure, so that it the legend fits to the figure.

In several places, "**wildtype**" must be corrected to "wild type".

Wildtype was changed to wild type throughout the text.

p. 10, line 268: The authors should add a sentence about the identification of the secondary metabolites and in addition they could refer to Table S6.

In the method section about secondary metabolites was now added that the secondary metabolites identified in this study are listed in table S10 (Line 729-730).

p. 12, line 327: How were the Western blot signals detected and quantified?

Information about the detection of the signals from the western experiments and their quantification was added to line 789-794.

p. 33, line 831: The authors could also shortly mention and discuss the regulator VadA in this context.

We stated in line 486-491 the role of the regulator VadA in the context of sporulation and its activation by the VelB-VosA heterodimer.

p. 21, Table 1: MS/MS counts is a less common term and should be explained or replaced by "Peptide Spectrum Matches". Or do the authors mean something different?

MS/MS count is now explained under the table in line 1259-1260 that is refers to the number of specific peptides fragmented and analyzed.

Hopefully, all points were covered adequate.

We are looking forward to hearing from you.

Sincerely yours,

Gerhard Braus

October 31, 2025

RE: Life Science Alliance Manuscript #LSA-2025-03395-TRR

Prof. Gerhard H Braus
University of Göttingen
Institute of Microbiology & Genetics
Molecular Microbiology and Genetics
Grisebachstr. 8
Göttingen D-37077
Germany

Dear Dr. Braus,

Thank you for submitting your Research Article entitled "The VelB IDD promotes selective heterodimer formation of velvet proteins for fungal development". It is a pleasure to let you know that your manuscript is now accepted for publication in Life Science Alliance. Congratulations on this interesting work.

During manuscript proofing, we recommend that you edit the abstract to more clearly delineate background information from the novel findings made in this work.

DISTRIBUTION OF MATERIALS:

Again, congratulations on a very nice paper. I hope you found the review process to be constructive and are pleased with how the manuscript was handled editorially. We look forward to future exciting submissions from your lab.

Sincerely,
